# Antiinflammation Derived Suzuki-Coupled Fenbufens as COX-2 Inhibitors: Minilibrary Construction and Bioassay

**DOI:** 10.3390/molecules27092850

**Published:** 2022-04-29

**Authors:** Shiou-Shiow Farn, Yen-Buo Lai, Kuo-Fong Hua, Hsiang-Ping Chen, Tzu-Yi Yu, Sheng-Nan Lo, Li-Hsin Shen, Rong-Jiun Sheu, Chung-Shan Yu

**Affiliations:** 1Department of Biomedical Engineering and Environmental Sciences, National Tsinghua University, Hsinchu 30013, Taiwan; amanda@iner.gov.tw (S.-S.F.); x74120robert@gmail.com (Y.-B.L.); s109012467@m109.nthu.edu.tw (H.-P.C.); s109012079@m109.nthu.edu.tw (T.-Y.Y.); lisinshen.nthu@gmail.com (L.-H.S.); 2Isotope Application Division, Institute of Nuclear Energy Research, Taoyuan 32546, Taiwan; loshengnan@iner.gov.tw; 3Department of Biotechnology and Animal Science, National Ilan University, Ilan 26007, Taiwan; kuofenghua@gmail.com; 4Institute of Nuclear Engineering and Science, National Tsinghua University, Hsinchu 30013, Taiwan; rjsheu@mx.nthu.edu.tw

**Keywords:** COX-2 selectivity, synergistic, COVID-19, biaryl, inflammasome

## Abstract

A small fenbufen library comprising 18 compounds was prepared via Suzuki Miyara coupling. The five-step preparations deliver 9–17% biphenyl compounds in total yield. These fenbufen analogs exert insignificant activity against the IL-1 release as well as inhibiting cyclooxygenase 2 considerably. Both the *para*-amino and *para*-hydroxy mono substituents display the most substantial COX-2 inhibition, particularly the latter one showing a comparable activity as celecoxib. The most COX-2 selective and bioactive disubstituted compound encompasses one electron-withdrawing methyl and one electron-donating fluoro groups in one arene. COX-2 is selective but not COX-2 to bioactive compounds that contain both two electron-withdrawing groups; disubstituted analogs with both resonance-formable electron-donating dihydroxy groups display high COX-2 activity but inferior COX-2 selectivity. In silico simulation and modeling for three COX-2 active—*p*-fluoro, *p*-hydroxy and *p*-amino—fenbufens show a preferable docking to COX-2 than COX-1. The most stabilization by the *p*-hydroxy fenbufen with COX-2 predicted by theoretical simulation is consistent with its prominent COX-2 inhibition resulting from experiments.

## 1. Introduction

Nonsteroidal anti-inflammatory drugs (NSAIDs) are characterized as cyclooxygenase (COX) inhibitors. NSAIDs have recently received significant attention, predominantly due to its unidentified adverse effects on treatments of the COVID-19 pandemic. In the course of COVID-19 therapy, one of the NSAIDs, Ibuprofen, induces the overexpression of angiotensin converting enzyme (ACE-2) receptor which may enable the entrance of SARS coronaviruses into the host cells [1]. Thus, the World Health Organization discouraged the repurposing of COX inhibitor.

The rising concerns of the adverse effects have been challenged by the report by Ong et al. [2]. They performed a large randomized controlled trial for the infected patients to assess COX inhibitor NSAIDs and COXIBs as appropriate therapeutics or adjuvant drugs against COVID-19. The supporting drugs administered with the NSAID ‘Etoricoxib’ reduced the levels of Interleukin-6, thus requiring no noninvasive or invasive ventilation or transfer to the intensive care units. Interestingly, the supporting drugs administered with NSAIDs did not develop adverse effects typically found in the cyclooxygenase inhibition therapy, e.g., gastrointestinal or cardiovascular intricacies. NSAIDs, such as aspirin, indomethacin, diclofenac, and celecoxib, have provided a silver lining in adjuvant COVID-19 therapy [3]. This can be attributed to their roles in confusing SARS viral replication and the deactivation of inflammasome. Synergistic inhibition of H5N1 viral infection with representative antiviral drugs is also remarkable.

Inflammation can be mediated through various pathways. Among them, COX plays a pivotal role. COX enzymes are principally classified into three types, i.e., COX-1, COX-2 and COX-3. COX-1 is constantly expressed in most tissues and functions as a house-keeping enzyme whereas COX-2 enzyme is induced in stress conditions and overexpressed in inflammation sites [4,5]. The uncommon COX-3 is a COX-1 splice variant [6,7,8]. In the presence of COX, arachidonic acid (AA) was metabolized to form prostaglandins (PG) via two sequential steps: a preliminary oxidation to PGG2 by using a COX followed by reduction to a labile endoperoxide intermediate PGH2 by peroxidase (POX). COXs have been treated with inhibitor NSAIDs but the mechanism remains complicated because AA and the inhibitor interact with COX reciprocally. COX-1 and COX-2 are also less frequently named as prostaglandin endoperoxide synthase-1 and -2 (PGHS-1 and -2) in the literature [9]. PGHSs are formed by two monomers with a physically distinct COX and POX active site, respectively. While the two monomers bear different conformations, they function cooperatively during catalysis in solution. The half-of-sites binding to one monomer may cause the partner monomer to undergo a conformational change. This may in turn act to allosterically modulate cyclooxygenase catalysis in the partner monomer [10]. As such, inhibitors such as NSAIDs with different classes of structures may target COXs with respect to different levels of regulation of the activities of COX-1 and COX-2.

NSAIDs have not only been used to treat a wide variety of diseases with relatively limited side effects [11], but can also exert COX-independent activity [12]. Such an action may inhibit the production of proinflammatory cytokines [13]. For example, NLRP3 characterized as an inflammasome-forming pattern recognition receptor can induce a series of cascade mechanisms via activation and secretion of factors such as caspase-1. It leads to a final release of proinflammatory cytokine interleukin-1β (IL-1β) causing inflammatory storm and diseases. The NLRP3 inflammasome is associated with various disease formations including Alzheimer’s disease and atherosclerosis as well as type 2 diabetes. It therefore attracts attention for discovering potentially therapeutic inflammasome inhibitors. As an example, fenamate NSAIDs with a biaryl structure have been shown to inhibit IL-1β secretion [11]. Likewise, another biaryl compound, glyburide, with an intrachain and a sulfone group, also inhibits NLRP3 in spite of its main therapeutic indications toward type 2 diabetes [14].

Taken together, from the needs of both COX-2 and NLRP3 selective inhibitors against the threatening diseases, we are interested in generating the diversity of the fenbufen analogs [15,16,17,18]. Arising from NSAIDs with the biaryl ring structurally analogous to fenamate, we report here the construction of a derivatized biphenyl ring and their biological assays for inhibiting the activities of COX-2 and COX-1 and the release of IL-1β. More specifically, considering the structural elements of biphenyl, sulfone group and substituents on aromatic rings that are required for inhibiting both COX-2 and NLRP3, a facial construction of the biphenyl ring with halogen functionality and sulfone group on the aromatic ring would be a key to bridging the bioorganic chemistry in this field.

## 2. Results and Discussion

The previously described *para*-iodofenbufen is not a rational candidate for diversifying the biphenyl scaffold (Figure 1) [15]. On account of the multi-step syntheses required for introducing various substituents, an alternative route toward the biphenyl ring needs to be conceived.

A facial method is mediated through Suzuki Miyara (SM) coupling of one benzene ring that is derivatized from commercial source or self-preparation. The corresponding fenbufen derivatives **5a**–**5o** were generated as shown in Figure 1. Starting from bromobenzene via Friedel–Crafts acylation to introduce side chain **2**, followed by borylation **3 [19]**, deprotection and the final SM coupling with the bromoarene derivatives, 15 fenbufen analogs can be generated [20,21,22,23,24,25]. Followed by further deprotection of the acid moiety, the final products **6a**–**6o** were obtained [21,26,27,28,29].

The above reactions are all performable with a rational total yield of 2–19% in five steps. Hydrolysis of the boronic ester **3** is a reversible reaction [30]. Although both stoichiometric amount and catalytic amount of 1 N HCl_(aq)_ have been reported, only catalytic amount is capable of delivering an optimized yield in our hand. The byproduct pinacol is easily removed and the crude product after extraction can be forwarded to the subsequent reaction.

SM coupling is a well-known carbon-carbon formation reaction. It can be affected by the catalyst, base and solvent. The combination of Pd(PPh_3_)_4_ or Pd(PPh_3_)_2_Cl_2_, Na_2_CO_3_ and dimethyloxyethane (DME) is mostly utilized [31]. When coupling the boronic acid **4** and the bromo counterpart according to the common reaction condition, only 6% yield of fenbufen analog **5d** was obtained (Table 1, entry 1). When trying other solvent and catalyst (entry 2) [32], we could merely observe the formation of the desired product **5d** at 4 h but it faded away at 18.5 h post the reaction along with the presence of a more concentrated polar unknown byproduct. Even after an attempt to shorten the reaction time, the yield remains unsatisfactory (entry 3 and 4).

Miyaura reported that basic condition may saponify the ester group, racemize chiral compounds and disable the condensation between aldehyde and alcohol [33]. Ester can be kept intact by employing a heterogeneous condition, such as the combination of aqueous K_2_CO_3_ and toluene, or the solid of K_3_PO_4_·nH_2_O or K_2_CO_3_ in toluene. While trying the condition of K_2_CO_3_, toluene and PdCl_2_ [34], reaction did not take place until the addition of a small amount of EtOH under assistance of a gradual increase of temperature to 75 °C from 90 min to 100 min post the reaction and a prolonged stirring overnight (entry 5). The isolated product fraction from flash chromatography was purified with HPLC to afford two fractions which were identified as a methyl ester **5b** and an ethyl ester **5bbyp** at a ratio of 1:1. The transesterification was not observed at the beginning of the reaction but it was observable after a cook. Thus, to prevent S-M coupling from the thermodynamic predominant transesterification, it needs to be stopped at 70 °C at 30 min post reaction in spite of a small amount of starting material remaining (entry 6). When substituting MeOH for EtOH, although the reaction proceeded rather fast, the yield did not improve significantly (entry 7). The optimized condition was met without transesterification when carried out at rt for 3 h (entry 8).

The SM coupling of the 1,2,4-tribromobenzene with organoboronic acid **4** generated three classes of products: tri-, di- and the major mono-coupled products. The fraction of mono coupled product mixtures from column chromatography could be further separated through HPLC to give three fractions at a yield ratio of 30:14:24 (Figure 2). As expected, the most steric hinderance C-2 decreases the yield of **5h**. Compared to Pd(PPh_3_)_4_, the less bulky catalyst PdCl_2_ assist the SM coupling to all three positions [35]. The three compounds are distinguishable using **^1^H-NMR** spectroscopy by comparing the deshielding effects arisen from the closeness to the two bromo groups [36].

In the course of the coupling, a homocoupled byproduct was always observed from TLC [30]. The byproduct did not really disturb the purification except the compound **5o**. The polar NH_2_ group forms a hydrogen bond with SiO_2_. NEt_3_ as a common co-eluent for chromatography will render the current mobile phases inadequate because of deterioration of the theoretical plates. Only when substituting EtOAc/CH_2_Cl_2_/NEt_3_ or CH_3_OH/CH_2_Cl_2_/NEt_3_ with acetone/*n*-hexane/NEt_3_ = 2:8:0.3, a rough purification was allowed. However, the solubility remains an unresolved issue even after adding CHCl_3_; the mixture will gradually precipitate in the column chromatography resulting in blockade. Nevertheless, the homocoupled byproduct could be removed in the next tosylation. The final deprotection of the acid group could be accomplished using CF_3_COOH at 120 °C whereas NaOH in CH_3_OH (aq) is a common condition [19,37].

In view of the contribution of sulfonamide and sulfonylurea to the pharmacophore for inhibiting NLRP3 [13,38], e.g., MCC950 and glyburide [39,40], we therefore introduced a tolylsulfonyl group to mimic the pharmacophore (Figure 2). Followed by tosylation and acid deprotection, the tosylate **8** was assessed for its bioactivity. To prevent the deprotected product from forming a TFA-co-crystalized complex that may alter the bioassay, LiOH was employed instead.

The following compounds are also included in the bioassay (Figure 3). Compound **9** [41] and **10** were each obtained from the ester congeners that had been reported before [19]. Furthermore, compound **11** was obtained from its ester precursor **12** that was prepared in a similar S-M coupling. In contrast to the present organoboron laying on the arene already installed with a carboxylic acid group, the corresponding boron building block for compound **12** is on the other benzene moiety.

As shown in Figure 4, all the fenbufen analogs do not inhibit the release of IL-1 to a satisfactory level. The two rigid biaryl rings without a heteroatom to link between them may hamper the activity as evidenced from the NLRP-3 potent flufenamic acid encompassing an azo between two arenes [13]. The COX inhibition was assessed using the commercial assay kit Cayman (No. 560131). The whole assay procedure is divided into two parts: COX inhibition and ELISA staining. The detection by an enzyme-linked immune assay (ELISA) is aimed at comparing 20 compounds tested to the standard COX-2 selective celecoxib and COX-1 selective resveratrol in a qualitative manner. To assess their bioactivities simultaneously, the procedure was modified to fit the requirements for a minimal volume of 20 μL by each multipipetting. Visible light absorbance of both the groups of void COX and fully active COX are well within the meaningful ranges guided by the assay kit. In some batches of experiments, the reaction tube embedded in a holder makes heat transfer inefficient. Thus, the reaction temperature may be lower than the optimal 37 °C rendering the reaction incomplete, thus voiding the result. Hence, the current data is grouped on the basis of a comparison of compounds using concentration of 22 μM (Figure 5, Figure 6 and Figure 7).

In general, all these fenbufen compounds are COX-2 selective (COX-2/COX-1 > 3). Some of them also show comparable COX-2 inhibition to that of celecoxib, such as the monosubstituted analogs *para*-fluoro **6a**, *p*-hydroxy **6l** and *p*-amino **6o** analogs. The former two compounds show 8-fold inhibition of COX-2 compared to COX-1, whereas **6o** shows 60-fold inhibition of COX-2 compared to COX-1. Concerning the disubstituted analogs, in spite of the high COX-2 selectivities by **6b**, **6g**, **6i** and **6j**, only **6b** exhibits an acceptable COX-2 inhibition. Compound **6b** contains one *ortho*-methyl as an electron-donating group and one *para*-fluoro as an electron-withdrawing group. The other three isosteres enclose both electron-withdrawing 4th periodic bromo groups. In addition, when encompassing two resonance-formable electron-donating groups, such as *o*- and *p*-dihydroxy compound **6m**, the COX-2 selectivity diminishes while COX-2 activity is reasonable.

Because of the substantial bioactivities of *p*-fluoro, *p*-hydroxy and *p*-amino fenbufen compounds **6a**, **6l**, **6o**, a further docking study was performed using in silico simulation and modeling. Two enzyme-sequence templates were retrieved from PDB bank encompassing COX-1 (1EQG) complexation with ibuprofen, a nonselective NSAID [42] and COX-2 (1CX2) complexation with SC558 (bromocelecoxib) [43], a COX-2 specific inhibitor. Before the molecular docking (MD) simulation, the receptor moieties, i.e., the two COX enzymes were both administered in the CHARMm force field throughout the whole docking process.

The flexible receptor atom property is enabled by creating a sphere radiating from a center defined by the PDB crystal data with a radius of 4 Ǻ (Section 4.4.1) [42,43]. The two sites contain mostly involved residues including Arg120 and Tyr355 for COX-1 and His90, Gln192, Arg513, Ser353, Tyr355 and Phe518 for COX-2.

The in situ ligand minimization algorithm comprises a number of programmings, such as adopted basis Newton–Raphson (NR), steepest descent and conjugate gradient. NR is applied to a subspace of the coordinate vector spanned by the displacement coordinates of the last positions. Steepest descent and conjugate gradient are both used to improve a poor conformation through an iterative method via minimization steps as well as the current gradient to determine the next step. Energy minimization through these procedures will be scored in terms of the function of smart minimizer.

A further algorithm for generating conformations was enabled by adopting the option of FAST mode so that rational numbers of low-energy conformation were obtained at a reasonable of time cost. The entropy component for the ligand conformation was also optimized.

Because the solvent effect plays an important role in the binding calculation, an implicit solvent model was performed with respect to Coulomb repulsion and dielectric attraction using the mode of Poisson–Boltzmann with non-polar surface area (PBSA). PBSA is the most rigorous yet slowest solvent approximation method based on continuum electrostatics. A further scoring function, such as the salt concentration, was also addressed. For example, NaCl as the salt and concentration was set to be 0.145 M.

Through a preliminary flexible docking of COX-1 and COX-2, the fluoro analog **6a** generated 36 and 30 docking poses, respectively; the hydroxy analog **6l** generated 56 and 96 poses for each; the amino analog **6o** provided 56 and 62 poses for each. According to the free energy derivation: ΔG_binding_ = ΔG_complex_ − ΔG_ligand_ − G_enzyme_, the top five high-scoring poses of each group were included in the binding free energy calculation and an average of the five data of each set are grouped in the Table 2. An exception is the fifth data of 5-hydroxy fenbufen **5l** docked to COX-2 showing an extraordinarily large value which is skipped. 

The current simulation was validated by redocking the benchmarking inhibitor ibuprofen to compare with the pose from the original crystallized COX-1 complex (1QEG) in the experimental (Section 4.4.4). The root-mean-square deviation (RMSD) value of 5.649 Å is larger than in the literature [44]. Similar findings were also observed in the case of redocking of celecoxib in comparison with the crystallized pose of bromo analog (SC558) complexed with COX-2 (1CX2). Whereas the electrondonating methyl group is virtually different from the electronwithdrawing bromo group, the RMSD value of 6.615 Å is comparable to the example of ibuprofen.

The values of free binding energy (ΔG_binding-CHARMm_) from the first calculation ranging from −60 to −250 kcal/mol is smaller than the reported values of around −10 kcal/mol [44,45]. Calibration by incorporating the solvent term provides the second set of data (ΔG_binding-MMPBSA_). Whereas the ΔG_binding-MMPBSA_ of three compounds (−13~−35 kcal/mol) are comparable to the literature, the two most COX-2 inhibiting *p*-hydroxy fenbufen **6l** and celecoxib scoring poor in ΔG_binding-MMPBSA_ = 23 kcal/mol seems to deviate unusually. The results may be due partly to the erroneous input of the original crystallized data of the 1eqg (COX-1) and 1CX2 (COX-2) at resolutions of 2.6 and 3.0 Å, respectively. The preliminary Gibbs free binding energy data under CHARMm condition provided a more consistent trend and formed the basis for comparison. It is noted that the COX-2 benchmarking inhibitor celecoxib scores are very low (ΔG_binding-CHARMm_ = −60 kcal/mol) compared with fenbufen analogs (−156 to −248 kcal/mol). Fenbufen analogs may take advantage through the fitting feasibility of linear-like structural flexibility compared with the relatively rigid tri-cyclic structure of celecoxib.

The MD simulations of fluoro analog **6a** with COX-1 and COX-2 show that both the long and narrow channels of the two active sites can accommodate fluoro analog **6a** (Figure 8 and Figure 9). Similar trends are also observed for the hydroxy and amino analogs **6l**, **6o** but are akin to bind more deeply in the active sites of COX-2 (Appendix A). The free binding energy values of fluoro analog **6a** to COX-2 are smaller but larger when docking to COX-1. This is also applicable to both the hydroxy and amino analogs **6l** and **6o**. Whereas van der Waals contacts between COX-1 and fluoro analog **6a** constitute the major stabilization, unfavorable interactions are also emerged and even more than that of COX-2’s docking. Hydroxy analog **6l** attains stabilization with both dockings of COX-1 and COX-2 in all respects of attractive interaction, such as the van der Waals and electrostatic interactions. Similar interaction patterns are observed in the case of amino analog **6o**. The RMSD values for hydroxy and amino analogs **6l**, **6o** coupled with both enzymes are within reasonable ranges (COX-1 and COX2; 0.91 Å and 1.22 Å; 1.59 Å and 1.25 Å) except fluoro analog **6a**, which shows relatively resonant values when docking to COX-1 (2.9 Å) but preserves reasonable values when docking to COX-2 (1.95 Å). In brief, the MD simulation results imply that the three compounds prefer a binding toward COX-2 than COX-1.

As shown in Table 2, while COX-2 inhibition by the benchmarking inhibitor celecoxib is comparable to the three fenbufen compounds, the stabilization predicted from the MD simulation is significantly less than that of the three fenbufen analogs. The mismatch may be caused by the fitting ability in terms of the geometric restrictions as described above. Nevertheless, the comparison among the three fenbufen analogs shows a dependency of the COX-2 activity on the MD simulation. For example, *p*-hydroxy fenbufen **6l** scores the highest stabilization and exerts the most COX-2 inhibition. On the other hand, a similar finding was also observed in the group of *p*-fluoro and *p*-amino fenbufen analogs **6a**, **6o** but with an inverted order, probably due to the very close scoring and the close COX-2 inhibiting efficacy. As shown in Figure 10, the superimposition of *p*-hydroxy fenbufen **6l** and COX-2 inhibitor bromocelecoxib (SC558) arranged themselves in a similar spatial orientation with a similar binding mode prevailing. Whereas the OH group of compound **6l** exerts a prominent H-bonding to the OH group of Tyr385; the bromo group of SC558 lacks the corresponding interaction. The van del Waals contact between the aromatic ring of **6l** and the nonpolar residues of Val 349 and Ala 527 enhances the stabilization. SC558 exerts a similar interaction but with extra stabilization through the two benzene rings. In addition, the polar groups of Arg 120, Arg 513 and His 90 can engage in the dipole-dipole attractions. The polar sulfone group of SC558 exerts similar interactions. In spite of the structural difference, they both follow a similar binding pattern which may address their equivalent COX-2 inhibition if the difference between bromo and methyl group in SC558 and celecoxib, respectively, could be neglected.

## 3. Conclusions

Through the Suzuki-Miyamura coupling reaction, a small library comprising 18 fenbufen compounds was generated. The optimized condition using PdCl_2_ and the cosolvents of toluene and EtOH can generally provide the coupled biphenyl products within 3 h. In spite of insignificant IL-1β inhibition activity, the NSAIDs derivatives show a typical COX inhibition. As anticipated, the NLRP-3 activity might be improved by introducing a linker such as an azo or a sulfide group. Among them, *p*-fluoro, *p*-hydroxyl and *p*-amino fenbufen analogs **6a**, **6l**, **6o** are better COX-2 inhibitors. The *p*-hydroxy fenbufen **6l** is even better than celecoxib at the concentration level of 22 μM at COX-2 inhibition. All these potential fenbufen analogs exert a better inhibition against COX-2 than COX-1. The COX-2 potent and highly COX-2 selective *p*-amino fenbufen **6o** needs to be further assessed for their inflammatory efficacy in vivo. The *p*-hydroxy fenbufen **6l** showing the most COX-2 inhibition also deserves further study. Sulfone analog **8** shows potential because of having minor NLRP3 activity, a remarkable COX-2 activity and a considerable COX-2 selectivity.

MD simulation using CHARMm force field along with the parameter settings and validation of the program using benchmarking COX-2 inhibitor bromocelecoxib (SC85) and COX-1 inhibitor ibuprofen generates two classes of binding free energy scoring expression with respect to the solvent effects. Whereas the data from incorporating solvent effects biases the results, the preliminary CHARMm-derived data exerts a consistent trend and forms the basis for discussion. The relatively higher stabilization gained by the current simulation (−70 to −190 kcal) than the reported data (~−10 kcal/mol) may be due to the erroneous input and structural features. The former arises from fair crystallographic resolution of the model systems 1QEG and 1CX2 (2.6 Å and 3.0 Å). The latter is related to the more feasibly structural flexibility exerted by the linear-like structural fenbufen than the tricyclic celecoxib analogs. The lowest free binding energy from the MD simulation by *p*-hydroxy fenbufen **6l** was consistent with its highest COX-2 inhibiting activity. In addition, *p*-fluoro and *p*-amino fenbufen analogs **6a**, **6o** exert comparable experimental COX-2 inhibitions which are consistent with their equivalent free binding energies as predicted by ΔG_binding-CHARMm_.

Both *p*-hydroxy fenbufen **6l** and bromocelecoxib (SC558) follow the similar binding pattern to COX-2, irrespective of the difference between methyl group and bromo group.

## 4. Materials and Methods

### 4.1. General Information

Most reagents and solvents were purchased from Fluka (St. Louis, MO, USA), Sigma-Aldrich (St. Louis, MO, USA), Alfa (Binfield, Berkshire, UK), Acros (Geal, Belgium), Showa (Tokyo, Japan) or TCI (Tokyo, Japan). These compounds were performed in dried glassware under a purge of nitrogen at room temperature unless otherwise noted. CH_2_Cl_2_ and toluene were dried over CaH_2_. CH_3_OH was dried over Mg and distilled prior to reactions. THF was treated with FeSO_4_⋅5H_2_O and dried over KOH followed by filtration and distilled over Na. DMSO and DMF were distilled over CaH_2_ under reduced pressure. NEt_3_ and pyridine were distilled over CaH_2_. The related reagents and solvents were obtained in reagent grade. The eluents for flash chromatography (e.g., EtOAc, acetone, and *n*-hexane) were of industrial grade. They were distilled prior to use. CHCl_3_ and CH_3_OH were of reagent grade and used without purification. NMR spectroscopy including **^1^H-NMR** (500 MHz) and ^13^C-NMR (125 MHz, DEPT-135) was performed by using a Unity Inova 500 MHz instrument (Varian, USA). Deuterated-solvents including CDCl_3_, CD_3_OD, C_6_D_6_ and DMSO-*d*_6_ were purchased from Aldrich (St. Louis, MO, USA). Low-resolution mass spectrometry (LRMS) was carried out on an ESI-MS spectrometer using a Varian 901-MS Liquid Chromatography Tandem Mass Q-TOF Spectrometer at the Department of Chemistry of National Tsing-Hua University (NTHU). LRMS was also performed at the Department of Applied Chemistry of National Chiao-Tung University (NCTU). High-resolution mass spectrometry (HRMS) was carried out using a Varian HPLC (Prostar series ESI/APCI) system coupled with a Varian 901-MS (FT-ICR Mass) mass detector and a triple quadrupole setting. Thin layer chromatography (TLC) was performed with TLC silica gel 60 F_254_ pre-coated plates (Machery-Nagel, Dueren, Germany) to monitor the starting materials and products upon visualization under UV light (254 nm). TLC plates were staining with either ninhydrin or ceric ammonium molybdate under heating. Celite 545 was obtained from Macherey-Nagel Inc. (Dueren, Germany). Strong acid cation exchange resin (H^+^) was obtained from Amberlite IR-120. Column chromatography was performed using Silicycle 60 silica gel (60–200 mesh, Quebec City, QC, Canada) under a slight pressure. Melting points were measured with a MEL-TEMP instrument (Barnstead international, Dubuque, IA, USA) without correction.

Normal phase HPLC constitutes an Agilent isocratic 1100 pump that was connected by a UV-VIS detector (254 nm) and a column of ZORBAX SIL column (9.4 mm × 250 mm, 5 μm), a combination of EtOAc and *n*-hexane as the mobile phase at a flow rate of 3 mL/min. A Rheodyne injector with a loop of 0.5 mL was employed.

### 4.2. Chemical Preparation

*para*-Bromofenbufen methyl ester **2**



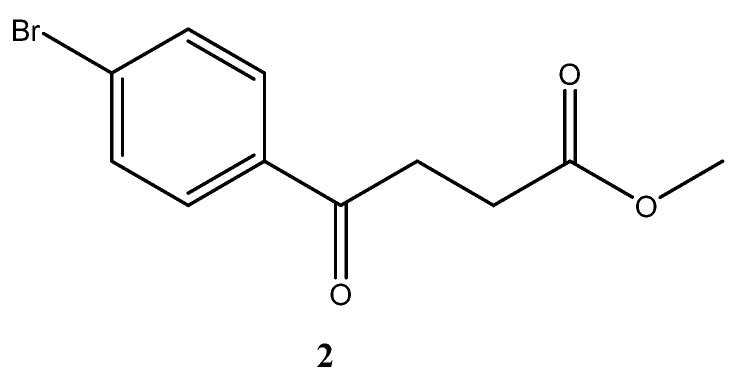



To a three-neck round bottomed flask was charged a mixture of succinic anhydride (31 g, 0.31 mol, 1.2 eq). CH_2_Cl_2_ (50 mL) was added and the mixture was stirred until dissolution. The mixture was cooled down by an ice bath followed by adding AlCl_3_ (104 g, 0.78 mol, 3 eq). The bath was removed and bromobenzene (27 mL, 0.26 mol) was added. CH_2_Cl_2_ (50 mL) was added and the sticky solid was agitated using a spatula. The mixture was poured into a mixture containing ice (120 g) and HCl (12 N, 65 mL) followed by Büchner filtration through suction. The white residue was further concentrated under reduced pressure. While attempting to recrystallize by using CH_3_OH for dissolution, a significant amount of methylated product was obtained but not to completeness. The mixture (35.5 g) after concentration under reduced pressure was submitted to an acid protection procedure as follows. The mixture was dried by distilling with toluene (10 mL) and CH_2_Cl_2_ (10 mL). CH_3_OH (100 mL) and H_2_SO_4_ (7.5 mL) was added. Stirring was allowed for 40 min followed by extraction using EtOAc (200 mL), Na_2_CO_3_ (satd., 50 mL × 2). The organic layers were collected and dried over Na_2_SO_4_ followed by gravitational filtration. The filtrate was concentrated under reduced pressure to give 49% yield of the white solid (35 g, 0.13 mol) over two steps (m.p. 48–50 °C (lit.[46] m.p. 51.5 °C).

**^1^H-NMR** (500 MHz, CDCl_3_) δ 2.70 (t, *J* = 6.5 Hz, 2H, H_Aliphatic_), 3.21 (t, *J* = 6.7 Hz, 2H, H_Aliphatic_), 3.64 (s, 3H, H_OCH3_), 7.54 (d, *J* = 8.0 Hz, 2H, H_Ar_), 7.78 (d, *J* = 8.5 Hz, 2H, H_Ar_); **^13^C-NMR** (125 MHz, CDCl_3_).

δ 27.92 (aliphatic, CH_2_), 33.33 (aliphatic, CH_2_), 51.88 (OCH_3_, CH_3_), 128.43 (Ar, C), 129.55 (Ar, CH), 131.95 (Ar, CH), 135.24 (Ar, C), 173.22 (CO-OCH_3_, C), 197.06 (Ar-CO, C).

methyl 4-oxo-4-(p-boronopinacoaryl)butanoate **3 [19]**



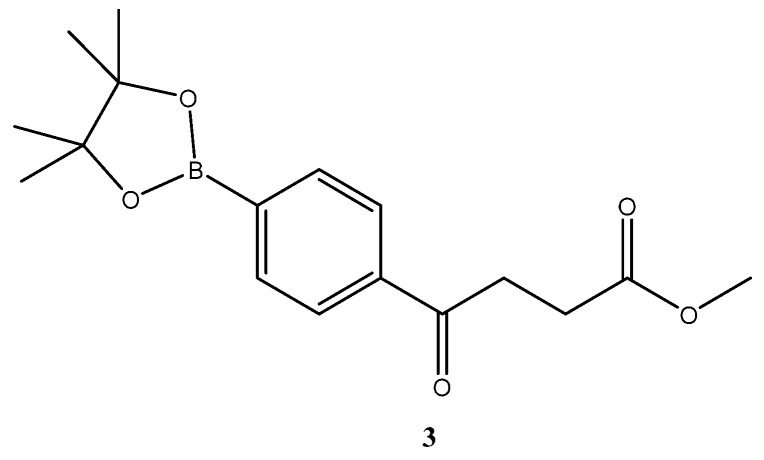



A two-necked round-bottomed flask (50 mL) encompassing KOAC (7.3 g, 73.8 mmol, 4.0 eq) was dried in an oven at 120 °C for over 96 h. A second flask (50 mL) charging diboronopinacol (9.4 g, 36.9 mmol, 2.0 eq) was dried by distillation with toluene at 50 °C under reduced pressure. Bromo compound **2** (5.0 g, 18.5 mmol, 1 eq) was added and co-distilled with toluene. A spin-like flask (50 mL) containing a complex of dichloro-[1,1′-bis(diphenylphosphino)ferrocene] palladium (II) [PdCl_2_(dppf)] (500 mg, 0.68 mmol, 10.0% wt) was dried twice with toluene by co-distillation at 55 °C under reduced pressure. A flask containing dried DMSO (over 4 Å MS for 48 h) was bubbled with N_2_ for 15 min. To the two-necked round bottomed flask containing KOAc, a mixture of bromo compound and diboronpinacol in DMSO (30 mL) and a solution of PdCl_2_(dppf) in DMSO (10 mL) were added sequentially under sufficient stirring. The mixture was moved to an oil bath that had been preheated to 90 °C and the stirring was continued through bubbling. The mixture turned from light orange to dark brown after 10 min. TLC (EtOAc/*n*-hexane 2:8) indicated the consumption of the starting material **2** (R*_f_* = 0.52) and formation of the product **3** (R*_f_* = 0.52) with intense blue after staining. The reaction was terminated at 110 min post reaction by partitioning between CH_2_Cl_2_ (40 mL) and HCl (aq. 0.2 N, 20 mL). The organic layer was dried over Mg_2_SO_4_ followed by filtration using a celite pad and concentration under reduced pressure. The black residue (25 g) contained a significant amount of DMSO that can be further reduced by a second extraction or purified through flash chromatography using the gradient mode of EtOAc/*n*-hexane 1:9 → 2:8 to give a pleasant Hinoki-essential oil-odor pale yellow viscous gum in a 78% yield (4.6 g).

**^1^H-NMR** (500 MHz, CDCl_3_) δ 1.33 (s, 12H, 4 × CH_3_), 2.75 (t, *J* = 6.5 Hz, 2H, -CH_2_), 3.31 (t, *J* = 6.5 Hz, 2H, -CH_2_), 3.68 (s, 3H, -OCH_3_), 7.87 (d, *J* = 8.0 Hz, 2H, aromatic), 7.93 (d, *J* = 8.0 Hz, 2H, aromatic); **^13^C NMR** (125 MHz, CDCl_3_) δ 24.86 (CH_3_, Bpin) 28.01 (CH_2_), 33.60 (CH_2_), 51.81 (CH_3_, O-CH_3_), 84.21 (C, Bpin), 126.99 (CH, aromatic), 134.94 (CH, aromatic), 138.40 (C, aromatic), 173.33 (C, COO), 198.31 (C, CO).

4-(4-methoxy-4-oxobutanoyl)phenyl)boronic acid **4**



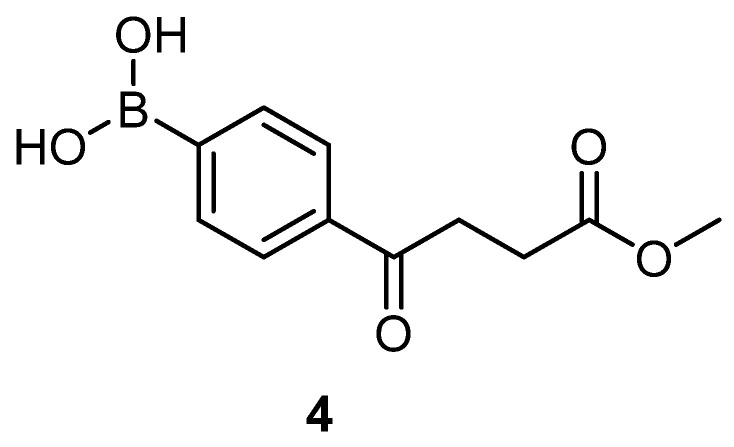



To a flask (100 mL) containing compound **3** (4.60 g, 14.4 mmol, 1.0 eq) was added THF (30 mL), H_2_O (7.5 mL) and NaIO_4_ (9.20 g, 43.2 mmol, 3.0 eq), sequentially. After 10 min, 1N HCl (1.5 mL, 1.5 mmol, 0.1 eq) was added. The white precipitate was stirred for 2 h to show a complete consumption of starting material (R*_f_* = 0.88) and formation of product (R*_f_* = 0.40) from TLC (acetone/*n*-hexane = 5:5). The mixture was partitioned between EtOAc (30 mL) and Na_2_CO_3_ (10 mL), followed by washing with sat. NaCl_(aq)_ (25 mL). The aqueous layers were further extracted with EtOAc (15 mL) × 3. The organic layers combined were dried over MgSO_4_ and filtered followed by concentration under reduced pressure to provide a white solid **4** (2.63 g, 77%). 148–153 °C.

**^1^H-NMR** (500 MHz, CDCl_3_/CD_3_OD = 1:1) δ 2.71 (t, *J* = 6.5 Hz, 2H, -CH_2_), 3.33 (t, *J* = 6.5 Hz, 2H, -CH_2_), 3.66 (s, 3H, -OCH_3_), 7.82 (d, 2H, aromatic), 7.92 (bs, *J* = 8.0 Hz, 2H, aromatic); **^13^C NMR** (125 MHz, CDCl_3_/CD_3_OD = 1:1) δ 28.79 (CH_2_), 33.38 (CH_2_), 52.20 (CH_3_, O-CH_3_), 127.82 (CH, aromatic), 134.92 (CH, aromatic), 138.79 (C, aromatic), 140.47 (C, bs, Ar-B(OH)_2_), 175.28 (C, COO), 200.54 (C, CO); analysis for C_11_H_13_BO_5_, calculated [M + Na]^+^ (*m*/*z*) = 259.0748 (100.0%), 258.0785 (24.8%), 260.0782 (9.7%), ESI-Q-TOF HR-ESI-MS found: [M + Na]^+^ = 259.0749 (46.6%), 258.0780 (11.2%), 260.0781 (5.8%), δ [ppm] = 0.4.

4-Boronopinacol phenylmethanol



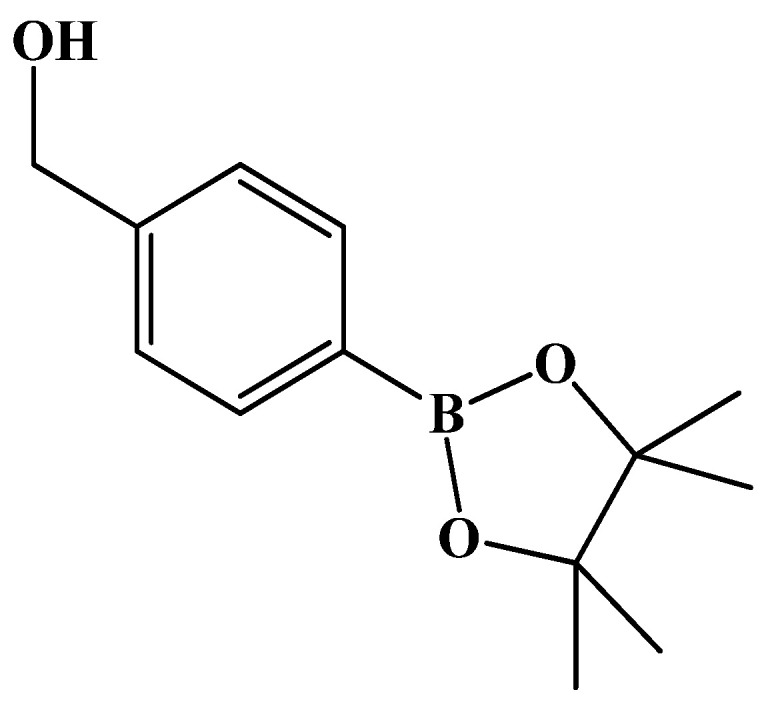



Reagents of commercial 4-bromophenyl methanol (2.0 g, 10.7 mmol, 1 eq), bis(pinacolato)diboron (5.4 g, 21.4 mmol, 2 eq), [PdCl_2_(dppf)] (200 mg, 10.0% wt) and KOAc (4.2 g, 42.8 mmol, 4 eq) as well as solvent DMF (20 mL) were used. Following the same procedure as that described for **3**, the reaction was performed under reflux for 4 h. The mixture was filtered followed by concentration under reduced pressure using an oil pump. The residue was extracted using EtOAc (30 mL) and saline (15 mL × 3). The organic layer was collected, dried over Na_2_SO_4_ and filtered through celite. After concentrating the filtrate under reduced pressure, the crude product (2.3 g) was chromatographed using gradient mode of EtOAc/*n*-hexane 2:8 → 4:6 to give a pale yellow solid in 81% yield (2 g). analysis for C_13_H_19_BO_3_ m.p. 63–65 °C (lit.[47] m.p. 62–64 °C white powder, lit.[48] oil).

**^1^H-NMR** (500 MHz, CDCl_3_) δ 1.35 (s, 12H, H_Methyl_), 4.72 (s, 2H, CH_2_), 7.37 (d, *J* = 7.5 Hz, 2H, H_Ar_), 7.80 (d, *J* = 8.0 Hz, 2H, H_Ar_), ^13^C-NMR (125 MHz, CDCl_3_) δ 24.85 (methyl, CH_3_), 65.29 (OH-CH_2_, CH_2_), 83.80 (BPin, C), 126.06 (Ar, CH), 135.05 (Ar, CH), 143.95 (Ar, C).

Methyl 4-(4′-chloro-[1,1′-biphenyl]-4-yl)-4-oxobutanoate **5d**



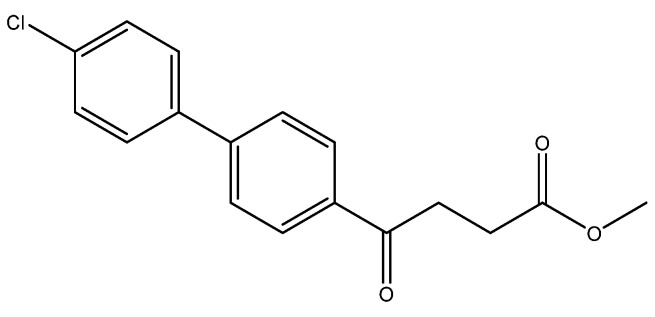



A flask charging K_2_CO_3_ (352 mg, 0.70 mmol, 3.1 eq) was dried at 120 °C for 48 h. The Erlenmeyer flask containing toluene (15 mL) and EtOH (15 mL) was bubbled thoroughly by N_2_ for 15 min. To the cooled flask of K_2_CO_3_ was added the bubbled cosolvents (3 mL), the mixture of **4** (200 mg, 0.85 mmol, 1.0 eq) and 1-bromo-4-chlorobenzene (326 mg, 1.70 mmol, 2.0 eq) in cosolvents (5 mL) of EtOH and toluene and PdCl_2_ (55 mg, 0.31 mmol, 6 mol%), sequentially. The mixture turned from red-brown to gray-black and finally to black in 1 h. The 3-h reaction indicated the consumption of the starting material **4** (R*_f_* = 0.20) and formation of the product (R*_f_* = 0.74) from TLC (acetone/*n*-hexane = 3/7). The mixture was extracted with 1 N HCl (0.2 mL), sat. NaCl_(aq)_ (1.5 mL) and H_2_O (5mL), sequentially. The organic layer was washed with CH_2_Cl_2_ (6 mL) and sat. NaCl_(aq)_ (1.5 mL). The organic layer along with the two organic layers used to back-extract the aqueous layer were dried over MgSO_4_ and filtered. After concentration of the filtrate under reduced pressure, the white residue (693.1 mg) was chromatographed using EtOAc/*n*-hexane in a gradient mode of 1/9 → 2/8 → 3/7 to provide the white solid **5d** in 77% yield (198 mg). m.p.: 117–120 °C.

**^1^H NMR** (500 MHz, CDCl_3_) δ 2.77 (t, *J* = 6.5 Hz, 2H, -CH_2_), 3.33 (t, *J* = 6.5 Hz, 2H, -CH_2_), 3.70 (s, 3H, -OCH_3_), 7.41 (dd, *J* = 7.0, 1.5 Hz, 2H, aromatic), 7.53 (dd, *J* = 7.0, 1.5 Hz, 2H, aromatic), 7.63 (d, *J* = 8.5 Hz, 2H, aromatic), 8.03 (d, *J* = 8.5 Hz, 2H, aromatic); **^13^C NMR** (125 MHz, CDCl_3_) δ 28.04 (CH_2_), 33.43 (CH_2_), 51.82 (CH_3_, O-CH_3_), 127.09 (CH, aromatic), 128.48 (CH, aromatic), 128.71 (CH, aromatic), 129.13 (CH, aromatic), 134.48 (C, aromatic), 135.52 (C, aromatic), 138.27 (C, aromatic), 144.58 (C, aromatic), 173.32 (C, COO), 197.50 (C, CO); analysis for C_17_H_15_ClO_3_, calculated [M + Na]^+^ (*m*/*z*) = 325.0602 (100.0%), 327.0572 (32.0%), 326.0635 (18.4%), 328.0606 (5.9%), ESI-Q-TOF HR-ESI-MS found: [M + Na]^+^ (*m*/*z*) = 325.0609 (100.0%), 327.0581 (30.9%), 326.0639 (15.9%), 328.0608 (5.1%), δ [ppm] = 2.1.

methyl 4-(4′-fluoro-[1,1′-biphenyl]-4-yl)-4-oxobutanoate **5a**



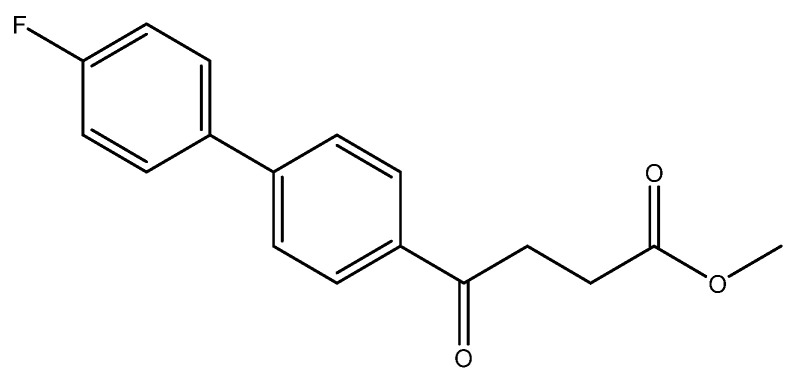



Reagents of **4** (52 mg, 0.21 mmol, 1.0 eq), bromo-4-fluorobenzene (73.5 mg, 0.42 mmol, 2.0 eq), K_2_CO_3_ (87 mg, 0.63 mmol, 3 eq) and PdCl_2_ (14.5 mg, 0.08 mmol, 6 mol%) and cosolvents of toluene/EtOH = 1:1 (3 mL) were used. Followed the procedure for that of **5d** for 3 h. TLC (acetone/*n*-hexane = 3/7) indicated the consumption of starting material (R*_f_* = 0.18) and formation of the product (R*_f_* = 0.52). The white solid of **5a** was obtained in 82% yield (50 mg, 0.17 mmol). m.p.: 127–130 °C.

**^1^H NMR** (500 MHz, CDCl_3_) δ 2.77 (t, *J* = 6.5 Hz, 2H, -CH_2_), 3.32 (t, *J* = 6.5 Hz, 2H, -CH_2_), 3.70 (s, 3H,-OCH_3_), 7.13 (ddd, *J* = 11.0, *J* = 5.0, *J* = 2.5 Hz, 2H, aromatic), 7.56 (ddd, *J* = 11.0, *J* = 5.0, *J* = 2.5 Hz, 2H, aromatic), 7.61 (d, *J* = 8.5 Hz, 2H, aromatic), 8.03 (d, *J* = 8.5 Hz, 2H, aromatic); **^13^C NMR** (125 MHz, CDCl_3_) δ28.03 (CH_2_), 33.34 (CH_2_), 51.80 (CH_3_, O-CH_3_), 115.88 (d, ^2^*J*_CF_ = 21.4 Hz, CH, aromatic), 127.07 (CH, aromatic), 128.66 (CH, aromatic), 128.89 (d, ^3^*J*_CF_ = 8.1 Hz, CH, aromatic), 135.24 (C, aromatic), 135.95 (d, ^4^*J*_CF_ = 2.8 Hz, C, aromatic), 144.83 (C, aromatic), 162.98 (d, ^1^*J*_CF_ = 248.4 Hz, C, aromatic) 173.33 (C, COO), 197.52 (C, CO); analysis for C_17_H_15_FO_3_, calculated [M + Na]^+^ (*m*/*z*) = 309.0903 (100.0%), 310.0931 (18.4%), 311.0965 (1.6%), [2M + Na]^+^ (*m*/*z*) = 595.1903 (100.0%), 596.1936 (36.8%), 597.1970 (6.6%), 597.1945 (1.2%), ESI-Q-TOF HR-ESI-MS found: [M + Na]^+^ = 309.0908 (75.0%), 310.0944 (14.2%), 311.0969 (1.22%), δ [ppm] = 1.8 [2M + Na]^+^ (*m*/*z*) = 595.1893 (100.0%), 596.1929 (36.8%), 597.1970 (37.2%), 597.1954 (8.9%), δ [ppm] = −1.7.

methyl 4-(4′-fluoro-2′-methyl-[1,1′-biphenyl]-4-yl)-4-oxobutanoate **5b**, ethyl 4-(4′-fluoro-2′-methyl-[1,1′-biphenyl]-4-yl)-4-oxobutanoate **5bbyp**



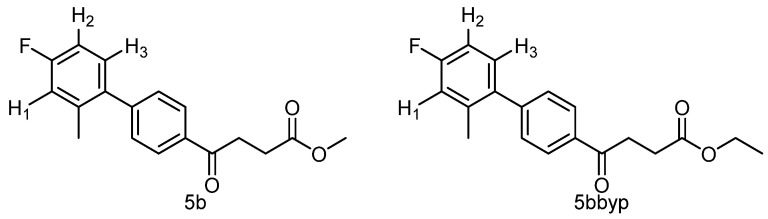



The reagents included **4** (52 mg, 0.21 mmol, 1.0 eq), 2-bromo-5-fluorotoluene (80 mg, 0.42 mmol, 2.0 eq), K_2_CO_3_ (87 mg, 0.63 mmol, 3 eq), PdCl_2_ (13 mg, 0.082 mmol, 6 mol%) and solvent of toluene (4 mL) and the procedure followed that of **5d**. The reaction post 1 h remained pale brown and TLC (acetone/*n*-hexane = 3:7) indicated no consumption of starting material (R*_f_* = 0.20). Additional EtOH (0.5 mL) was added and it turned gray and clear. After a further reaction at rt for 1.5 h, it was heated to 70–75 °C for 17.5 h. TLC (acetone/*n*-hexane = 3:7) indicated the formation of the two products (R*_f_* = 0.68, 0.70). After chromatography, a plastic smell and viscous liquid of the mixture of **5b** and **5bbyp** (40 mg, 62%). The mixture was purified using HPLC with isocratic condition of EtOAc/n-hexane = 1:9 to give the methyl fenbufen analog **5b** and ethyl fenbufen analog **5bbyp** in a ratio of 1:1. Another batch of experiment was optimized by using toluene/EtOH = 1:1 (3 mL) at 70 °C for 30 min. The viscous transparent liquid **5b** was obtained in 75% yield (47 mg). **5b** m.p.: 75–77 °C, **5bbyp** m.p.: 51–53 °C.

methyl 4-(4′-fluoro-2′-methyl-[1,1′-biphenyl]-4-yl)-4-oxobutanoate **5b**

**^1^H NMR** (500 MHz, CDCl_3_) δ 2.23 (s, 3H, Ar-CH_3_), 2.78 (t, *J* = 6.5 Hz, 2H, CH_2_), 3.34 (t, *J* = 6.5 Hz, 2H, CH_2_), 3.70 (s, 3H,-OCH_3_), 6.93 (t, *J* = 8.5, *J* = 2.5 Hz, 1H, aromatic), 6.97 (ddd, *J* = 9.5, *J* = 8.5, *J* = 2.5 Hz, 1H, aromatic), 7.15 (t, *J* = 8.5, *J* = 6.0 Hz, 1H, aromatic), 7.37 (d, *J* = 8.5 Hz, 2H, aromatic), 8.12 (d, *J* = 8.0 Hz, 2H, aromatic); **^13^C NMR** (125 MHz, CDCl_3_) δ 20.51 (Ar-CH_3_), 28.05 (CH_2_), 33.41 (CH_2_), 51.84 (CH_3_, O-CH_3_), 112.77 (d, ^2^*J*_CF_ = 21.2 Hz, CH, aromatic), 116.99 (d, ^2^*J*_CF_ = 21.2 Hz, CH, aromatic), 128.00 (CH, aromatic), 129.58 (CH, aromatic), 130.97 (d, ^3^*J*_CF_ = 8.5 Hz, CH, aromatic), 135.14 (C, aromatic), 136.72 (d, ^4^*J*_CF_ = 2.5 Hz, C, aromatic), 137.63 (d, ^3^*J*_CF_ = 7.9 Hz, C, aromatic), 146.12 (C, aromatic), 162.31 (d, ^1^*J*_CF_ = 245.1 Hz, C, aromatic) 173.37 (C, COO), 197.69 (C, CO); analysis for C_18_H_17_FO_3_, calculated [M + Na]^+^ (*m*/*z*) = 323.1054 (100.0%), 324.1087 (19.5%), 325.1121 (1.8%), ESI-Q-TOF HR-ESI-MS found: [M + Na]^+^ = 323.1058 (94.8%), 324.1086 (16.6%), 325.1115 (2.4%), δ [ppm] = 1.2.

ethyl 4-(4′-fluoro-2′-methyl-[1,1′-biphenyl]-4-yl)-4-oxobutanoate **5bbyp**

**^1^H NMR** (500 MHz, CDCl_3_) δ 1.26 (t, *J* = 7.0 Hz, 3H, CH_3_), 2.23 (s, 3H, Ar-CH_3_), 2.77 (t, *J* = 6.5 Hz, 2H, CH_2_), 3.33 (t, *J* = 6.5 Hz, 2H, CH_2_), 4.16 (q, *J* = 7.0 Hz, 2H,-OCH_2_), 6.93 (dd, *J*_HH_ = 8.5 Hz, *J*_HF_ = 2.5 Hz, 1H, aromatic), 6.97 (ddd, *J* = 9.5, *J* = 8.5, *J* = 2.5 Hz, 1H, aromatic), 7.15 (dd, *J* = 8.5, *J* = 6.0 Hz, 1H, aromatic), 7.37 (d, *J* = 8.5 Hz, 2H, aromatic), 8.01 (d, *J* = 8.0 Hz, 2H, aromatic); **^13^C NMR** (125 MHz, CDCl_3_) δ 14.19 (-CH_3_, O-CH_2_CH_3_), 20.51 (Ar-CH_3_), 28.33 (CH_2_), 33.41 (CH_2_), 60.68 (CH_2_, O-CH_2_CH_3_), 112.76 (d, ^2^*J*_CF_ = 20.9 Hz, CH, aromatic), 116.98 (d, ^2^*J*_CF_ = 21.3 Hz, CH, aromatic), 128.00 (CH, aromatic), 129.57 (CH, aromatic), 130.97 (d, ^3^*J*_CF_ = 8.5 Hz, CH, aromatic), 135.20 (C, aromatic), 136.73 (d, ^4^*J*_CF_ = 2.3 Hz, C, aromatic), 137.63 (d, ^3^*J*_CF_ = 7.9 Hz, C, aromatic), 146.07 (C, aromatic), 162.30 (d, ^1^*J*_CF_ = 245.1 Hz, C, aromatic) 172.92 (C, COO), 197.80 (C, CO); analysis for C_19_H_19_FO_3_, calculated [M + Na]^+^ (*m*/*z*) = 337.1210 (100.0%), 338.1244 (20.5%), 339.1278 (2.0%), ESI-Q-TOF HR-ESI-MS found: [M + Na]^+^ = 337.1209 (100.0%), 338.1236 (16.8%), 339.1268 (2.3%), δ [ppm] = −0.3.

methyl 4-(2′,4′-difluoro-[1,1′-biphenyl]-4-yl)-4-oxobutanoate **5c**



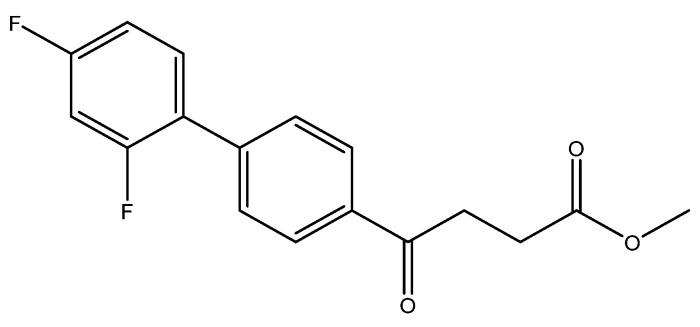



Reagents of **4** (200 mg, 0.85 mmol, 1 eq), 1-bromo-2, 4-difluorobenzene (328 mg, 1.70 mmol, 2 eq), K_2_CO_3_ (352 mg, 2.55 mmol, 3 eq) and PdCl_2_ (55 mg, 0.31 mmol, 6 mol%) and cosolvents of toluene/EtOH = 1:1 (7.5 mL) were used. Following the procedure for that of **5d** for 3 h. TLC (EtOAc/*n*-hexane = 4/6) indicated the consumption of starting material (R*_f_* = 0.20) and formation of the product (R*_f_* = 0.72). The white solid of **5c** was obtained in 69% yield (176 mg, 0.58 mmol). m.p.: 92–94 °C.

**^1^H NMR** (500 MHz, CDCl_3_) δ 2.78 (t, *J* = 6.5 Hz, 2H, CH_2_), 3.33 (t, *J* = 6.5 Hz, 2H, CH_2_), 3.70 (s, 3H, OCH_3_), 6.92 (td, *J* = 9.0, *J* = 2.5 Hz, 1H, aromatic), 6.96 (dd, *J* = 8.5, *J* = 2.5 Hz, 1H, aromatic), 7.41 (td, *J* = 8.5, *J* = 8.5, *J* = 6.0 Hz, 1H, aromatic), 7.59 (d, *J* = 8.5, *J* = 2.0 Hz, 2H, aromatic), 8.03 (d, *J* = 8.5 Hz, 2H, aromatic); **^13^C NMR** (125 MHz, CDCl_3_) δ 28.03 (CH_2_), 33.44 (CH_2_), 51.83 (CH_3_, O-CH_3_), 104.61 (t, ^2^*J*_CF_ = 25.8 Hz, CH, C-3′), 111.85 (dd, ^2^*J*_CF_ = 21.0 Hz, ^4^*J*_CF_ = 3.8 Hz, CH, C-5′), 124.18 (dd, ^2^*J*_CF_ = 13.4 Hz, ^4^*J*_CF_ = 3.8 Hz, C, C-1′) 128.27 (CH, aromatic), 129.10 (d, *J*_CF_ = 2.8 Hz, CH, aromatic), 131.39 (dd, ^3^*J*_CF_ = 9.6 Hz, ^3^*J*_CF_ = 4.6 Hz, CH, C-6′), 135.63 (C, aromatic), 139.80 (C, aromatic), 159.79 (dd, ^1^*J*_CF_ = 250.1 Hz, ^3^*J*_CF_ = 12.5 Hz, C-F, C-2′), 162.82 (dd, ^1^*J*_CF_ = 249.0 Hz, ^3^*J*_CF_ = 12.5 Hz, C-F, C-4′), 173.31 (C, COO), 197.55 (C, CO); analysis for C_17_H_14_F_2_O_3_, calculated [M + Na]^+^ (*m*/*z*) = 327.0803 (100.0%), 328.0837 (18.4%), 329.0870 (1.6%), ESI-Q-TOF HR-ESI-MS found: [M + Na]^+^ = 327.0802 (96.2%), 328.0835 (17.0%), 329.0868 (2.3%), δ [ppm] = 1.5.

methyl 4-(4′-chloro-2′-methyl-[1,1′-biphenyl]-4-yl)-4-oxobutanoate **5e**



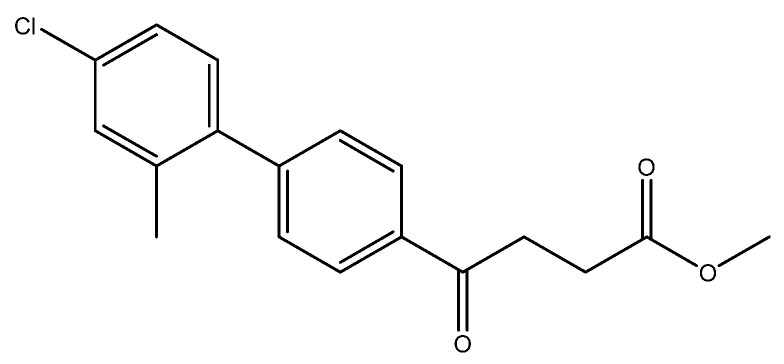



Reagents of **4** (52 mg, 0.21 mmol, 1.0 eq), 2-bromo-5-chlorotoluene (86 mg, 0.42 mmol, 2.0 eq), K_2_CO_3_ (96 mg, 0.70 mmol, 3.3 eq) and PdCl_2_ (13 mg, 0.082 mmol, 6 mol%) and cosolvents of toluene/CH_3_OH = 1:1 (3 mL) were used. Following the procedure for that of **5d** for 3 h. TLC (acetone/*n*-hexane = 3:7) indicated the consumption of starting material (R*_f_* = 0.20) and formation of the product (R*_f_* = 0.62). The white solid of **5e** was obtained in 71% yield (46 mg, 0.15 mmol). m.p.: 77–78 °C.

**^1^H NMR** (500 MHz, CDCl_3_) δ 2.22 (s, 3H, Ar-CH_3_), 2.78 (t, *J* = 6.5 Hz, 2H, CH_2_), 3.30 (t, *J* = 6.5 Hz, 2H, CH_2_), 3.70 (s, 3H, -OCH_3_), 7.12 (d, *J* = 8.5 Hz, 1H, aromatic), 7.21 (d, *J* = 8.5, *J* = 2.0 Hz, 1H, aromatic), 7.26 (s, 1H, aromatic), 7.37 (d, *J* = 8.5 Hz, 2H, aromatic), 8.01 (d, *J* = 8.5 Hz, 2H, aromatic); **^13^C NMR** (125 MHz, CDCl_3_) δ 20.25 (Ar-CH_3_), 28.02 (CH_2_), 33.40 (CH_2_), 51.81 (CH_3_, O-CH_3_), 126.02 (CH, aromatic), 128.02 (CH, aromatic), 129.39 (CH, aromatic), 130.32 (CH, aromatic), 130.68 (CH, aromatic), 133.59 (C, aromatic), 135.27 (C-Cl, aromatic), 137.08 (C, aromatic), 139.16 (C, aromatic), 145.83 (C, aromatic) 173.32 (C, COO), 197.63 (C, CO); analysis for C_18_H_17_ClO_3_, calculated [M + H]^+^ (*m*/*z*) = 317.0946 (100.0%), [M + Na]^+^ (*m*/*z*) = 339.0764 (100.0%), 341.0729 (32.0%), 340.0792 (19.5%), ESI-Q-TOF HR-ESI-MS found: [M + H]^+^ (*m*/*z*) = 317.0942 (12.7%), δ [ppm] = −0.8, [M + Na]^+^ = 339.0766 (100.0%), 341.0743 (38.5%), 340.0786 (25.9%), δ [ppm] = 0.6.

methyl 4-(2′,4′-dichloro-[1,1′-biphenyl]-4-yl)-4-oxobutanoate **5f**



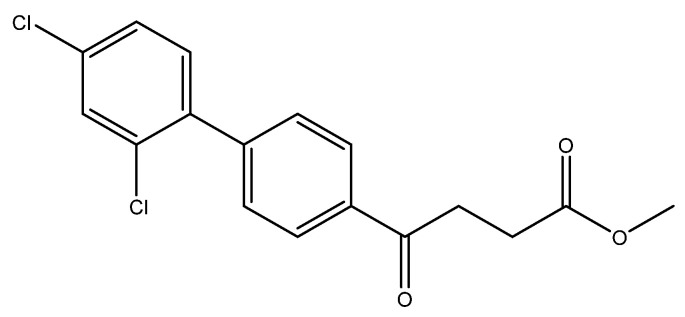



Reagents of **4** (200 mg, 0.85 mmol, 1 eq), 1-bromo-2, 4-dichlorobenzene (384 mg, 1.70 mmol, 2 eq), K_2_CO_3_ (352 mg, 2.55 mmol, 3 eq) and PdCl_2_ (55 mg, 0.31 mmol, 6 mol%) and cosolvents of toluene/EtOH = 1:1 (7.5 mL) were used. Following the procedure for that of **5d** for 27 h. TLC (EtOAc/*n*-hexane = 4/6) indicated the consumption of starting material (R*_f_* = 0.20) and formation of the product (R*_f_* = 0.76). The white solid of **5f** was obtained in 69% yield (197 mg, 0.58 mmol).

**^1^H NMR** (500 MHz, CDCl_3_) δ 2.73 (t, *J* = 6.5 Hz, 2H, CH_2_), 3.29 (t, *J* = 6.5 Hz, 2H, CH_2_), 3.65 (s, 3H, -OCH_3_), 7.20 (dd, *J* = 8.5, 2.0 Hz, 1H, aromatic), 7.26 (dd, *J* = 8.5, 2.0 Hz, 1H, aromatic), 7.45 (dd, *J* = 7.0, 2.0 Hz, 3H, aromatic), 7.98 (dd, *J* = 7.0, 2.0 Hz, 2H, aromatic); **^13^C NMR** (125 MHz, CDCl_3_) δ 28.00 (CH_2_), 33.46 (CH_2_), 51.85 (O-CH_3_), 127.33 (CH, aromatic), 127.94 (CH, aromatic), 129.69 (CH, aromatic), 129.91 (CH, aromatic), 131.81 (CH, aromatic), 133.08 (C-Cl, aromatic), 134.48 (C-Cl, aromatic), 135.80 (C, aromatic), 137.90 (C, aromatic), 143.09 (C, aromatic), 174.04 (C, COO), 198.44 (C, CO); analysis for C_17_H_14_Cl_2_O_3_, calculated [M + Na]^+^ (*m*/*z*) = 359.0212 (100.0%), 361.0183 (63.9%), 360.0246 (18.4%), 362.0216 (11.8%), 363.0153 (10.2%), ESI-Q-TOF HR-ESI-MS found: [M + Na]^+^ = 359.0212 (88.1%), 361.0185 (60.0%), 360.0250 (16.6%), 362.0220 (10.8%), 363.0158 (10.3%), δ [ppm] = 0.

methyl 4-(3′,4′-dibromo-[1,1′-biphenyl]-4-yl)-4-oxobutanoate **5g**, methyl 4-(2′,5′-dibromo-[1,1′-biphenyl]-4-yl)-4-oxobutanoate **5h**, methyl 4-(2′,4′-dibromo-[1,1′-biphenyl]-4-yl)-4-oxobutanoate **5i**



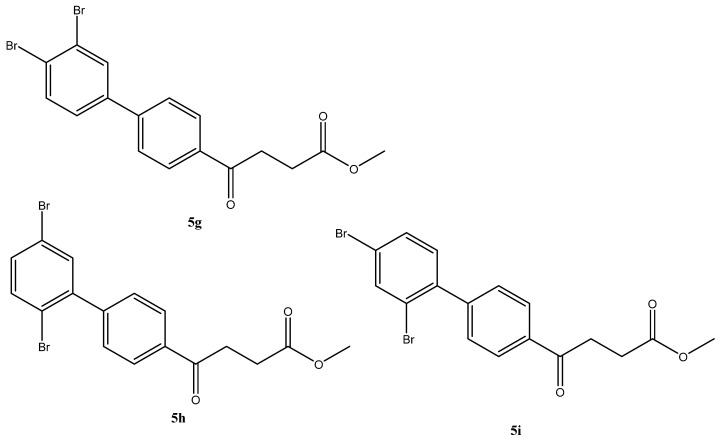



Reagents of **4** (200 mg, 0.85 mmol, 1 eq), 1, 2, 4-tribromobenzene (535 mg, 1.70 mmol, 2 eq), K_2_CO_3_ (352 mg, 2.55 mmol, 3 eq) and PdCl_2_ (55 mg, 0.31 mmol, 6 mol%) and cosolvents of toluene/EtOH = 1:1 (7.5 mL) were used. Following the procedure that for **5d** for 2 h. TLC (EtOAc/*n*-hexane = 3/7) indicated the consumption of starting material (R*_f_* = 0.20) and formation of the product (R*_f_* = 0.54–0.64). The white solid was obtained as a mixture in 68% yield (246 mg). A portion of the sample (50 mg) was analyzed using normal phase HPLC per isocratic mode with eluents of EtOAc/*n*-hexane = 1/10 to give white solid **5g,** viscous liquid **5h** and white solid **5i** in 30% (20 mg), 14% (9 mg) and 24% (16 mg) yield, respectively. Due to the limited amount of **5h**, spectroscopic measurement was not taken. It was directly deprotected for subsequent spectroscopic analysis and biological assay.

methyl 4-(3′,4′-dibromo-[1,1′-biphenyl]-4-yl)-4-oxobutanoate **5g**

**^1^H NMR** (500 MHz, CDCl_3_) δ 2.77 (t, *J* = 6.5 Hz, 2H, CH_2_), 3.32 (t, *J* = 6.5 Hz, 2H, CH_2_), 3.70 (s, 3H, -OCH_3_), 7.38 (d, *J*_5′,6′_ = 8.5 Hz, *J*_2′,5′_ = 2.0 Hz, 1H, aromatic), 7.61 (d, *J* = 8.5 Hz, 2H, aromatic), 7.68 (d, *J* = 8.5 Hz, 2H, aromatic), 7.84 (s, *J*_2′,5′_ = 2.0 Hz, 1H, aromatic), 8.03 (d, *J* = 8.5 Hz, 2H, aromatic); **^13^C NMR** (125 MHz, CDCl_3_) δ 28.00 (CH_2_), 33.47 (CH_2_), 51.83 (CH_3_, O-CH_3_), 124.78 (C-Br, aromatic), 125.47 (C-Br, aromatic), 127.10 (CH, aromatic), 127.21 (CH, aromatic), 128.78 (CH, aromatic), 132.23 (CH, aromatic), 134.07 (CH, aromatic), 135.97 (C, aromatic), 140.55 (C, aromatic), 143.18 (C, aromatic), 173.28 (C, COO), 197.42 (C, CO); analysis for C_17_H_14_Br_2_O_3_, calculated [M + Na]^+^ (*m*/*z*) = 448.9181 (100.0%), 446.9207 (51.4%), 450.9161 (48.6%), 449.9215 (9.7%), 451.9195 (8.9%), ESI-Q-TOF HR-ESI-MS found: [M + Na]^+^ = 448.9183 (100.0%), 446.9204 (78.7%), 450.9177 (81.2%), 449.9230 (32.8%), 451.9214 (1.3%), δ [ppm] = 0.4.

methyl 4-(2′,5′-dibromo-[1,1′-biphenyl]-4-yl)-4-oxobutanoate **5h**

**^1^H NMR** (500 MHz, CDCl_3_) δ 2.78 (t, *J* = 6.5 Hz, 2H, CH_2_), 3.34 (t, *J* = 6.5 Hz, 2H, CH_2_), 3.70 (s, 3H,-OCH_3_), 7.35 (d, *J*_5′,6′_ = 8.5 Hz, *J*_3′,5′_ = 2.5 Hz, 1H, aromatic), 7.44 (s, *J*_3′,5′_ = 2.5 Hz, 1H, aromatic), 7.47 (d, *J* = 8.5 Hz, 2H, aromatic), 7.52 (d, *J* = 8.5 Hz, 2H, aromatic), 8.03 (d, *J* = 8.5 Hz, 2H, aromatic); **^13^C NMR** (125 MHz, CDCl_3_) δ 28.01 (CH_2_), 33.50 (CH_2_), 51.85 (CH_3_, O-CH_3_), 120.94 (C-Br, aromatic), 121.32 (C-Br, aromatic), 127.93 (CH, aromatic), 129.60 (CH, aromatic), 132.30 (CH, aromatic), 133.66 (CH, aromatic), 134.61 (CH, aromatic), 135.99 (CH, aromatic), 143.25 (C, aromatic), 144.48 (C, aromatic), 173.30 (C, COO), 197.55 (C, CO); analysis for C_17_H_14_Br_2_O_3_, calculated [M + Na]^+^ (*m*/*z*) = 448.9181 (100.0%), 446.9207 (51.4%), 450.9161 (48.6%), 449.9215 (9.7%), 451.9195 (8.9%), 447.9235 (5.0%), ESI-Q-TOF HR-ESI-MS found: [M + Na]^+^ = 448.9187 (100.0%), 446.9207 (57.1%), 450.9177 (54.2%), 449.9205 (27.7%), 451.9360 (2.1%), 447.9314 (3.4%), δ [ppm] = 1.3.

methyl 4-(2′,4′-dibromo-[1,1′-biphenyl]-4-yl)-4-oxobutanoate **5i**

**^1^H NMR** (500 MHz, CDCl_3_) δ 2.78 (t, *J* = 6.5 Hz, 2H, -CH_2_), 3.38 (t, *J* = 6.5 Hz, 2H, CH_2_), 3.70 (s, 3H, OCH_3_), 7.17 (d, *J* = 8.0 Hz, 1H, aromatic), 7.46 (d, *J* = 8.0 Hz, 2H, aromatic) 7.50 (d, *J*_3′,4′_ = 8.0 Hz, *J*_4′,6′_ = 2.0 Hz, 1H, aromatic), 7.84 (s, *J*_4′,6′_ = 2.0 Hz, 1H, aromatic), 8.03 (d, *J* = 8.0 Hz, 2H, aromatic); **^13^C NMR** (125 MHz, CDCl_3_) δ 28.02 (CH_2_), 33.48 (CH_2_), 51.84 (CH_3_, O-CH_3_), 122.32 (C-Br, aromatic), 122.89 (C-Br, aromatic), 127.91 (CH, aromatic), 129.59 (CH, aromatic), 130.73 (CH, aromatic), 131.94 (CH, aromatic), 135.60 (CH, aromatic), 135.86 (CH, aromatic), 140.46 (C, aromatic), 144.78 (C, aromatic), 173.29 (C, COO), 197.56 (C, CO).

methyl 4-(3′,5′-dibromo-[1,1′-biphenyl]-4-yl)-4-oxobutanoate **5j**



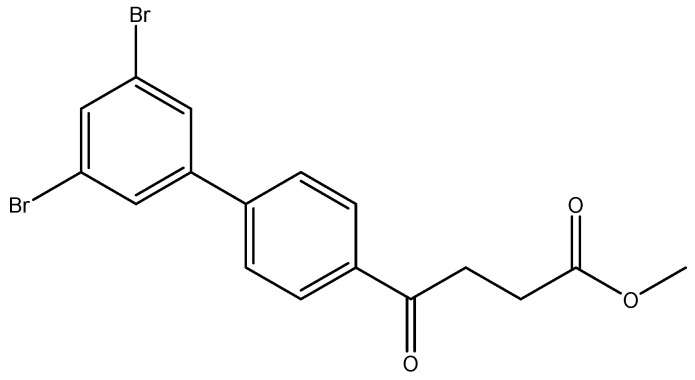



Reagents of **4** (200 mg, 0.85 mmol, 1 eq), 1-bromo-2, 4-difluorobenzene (538 mg, 1.70 mmol, 2 eq), K_2_CO_3_ (352 mg, 2.55 mmol, 3 eq) and PdCl_2_ (55 mg, 0.31 mmol, 6 mol%) and cosolvents of toluene/EtOH = 1:1 (7.5 mL) were used. Following the procedure for that of **5d** for 3 h. TLC (EtOAc/*n*-hexane = 4/6) indicated the consumption of starting material (R*_f_* = 0.40) and formation of the product (R*_f_* = 0.72). The white solid of **5j** was obtained in 36% yield (141 mg, 0.31 mmol). m.p.: 107–108 °C.

**^1^H NMR** (500 MHz, CDCl_3_) δ 2.78 (t, *J* = 6.5 Hz, 2H, CH_2_), 3.33 (t, *J* = 6.5 Hz, 2H, CH_2_), 3.70 (s, 3H,-OCH_3_), 7.60 (d, *J* = 8.5 Hz, 2H, aromatic), 7.66 (dd, *J* = 9.0, 1.0 Hz, 3H, aromatic), 8.04 (d, *J* = 8.5 Hz, 2H, aromatic); **^13^C NMR** (125 MHz, CDCl_3_) δ 28.00 (CH_2_), 33.50 (CH_2_), 51.86 (CH_3_, O-CH_3_), 123.47 (C-Br, aromatic), 127.33 (CH, aromatic), 128.78 (CH, aromatic), 129.09 (CH, aromatic), 133.52 (CH, aromatic), 136.24 (C, aromatic), 142.80 (C, aromatic), 143.39 (C, aromatic), 173.28 (C, COO), 197.41 (C, CO); analysis for C_17_H_14_Br_2_O_3_, calculated [M + Na]^+^ (*m*/*z*) = 448.9181 (100.0%), 446.9202 (51.4%), 450.9161 (48.6%), 449.9215 (9.7%), 451.9195 (8.9%), ESI-Q-TOF HR-ESI-MS found: [M + Na]^+^ = 448.9192 (25.8%), 446.9194 (13.7%), 450.9177 (12.8%), 449.9228 (4.8%), 451.9183 (2.8%), δ [ppm] = −2.4.

methyl 4-(4′-acetyl-[1,1′-biphenyl]-4-yl)-4-oxobutanoate **5k**



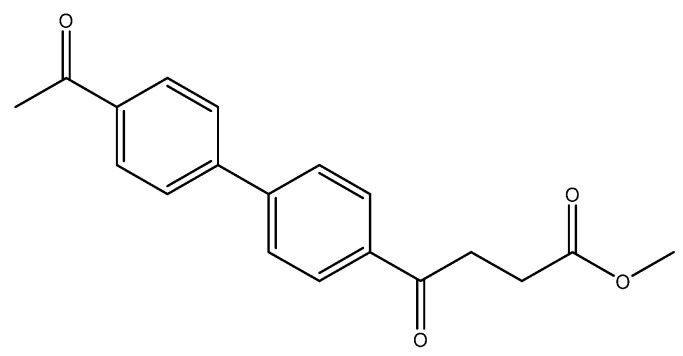



Reagents of **4** (200 mg, 0.85 mmol, 1 eq), 4-bromoacetophenone (338.4 mg, 1.70 mmol, 2 eq), K_2_CO_3_ (352 mg, 2.55 mmol, 3 eq) and PdCl_2_ (55 mg, 0.31 mmol, 6 mol%) and cosolvents of toluene/EtOH = 1:1 (7.5 mL) were used. Following the procedure for that of **5d** for 1 h. TLC (acetone/*n*-hexane = 3/7) indicated the consumption of starting material (R*_f_* = 0.18) and formation of the product (R*_f_* = 0.40). The mixture was chromatographed using eluents of CH_3_OH/CH_2_Cl_2_ = 1/99 to provide the white solid **5k** in 81% yield (215 mg, 0.69 mmol). 151–158 °C.

**^1^H NMR** (500 MHz, CDCl_3_) δ 2.62 (s, 3H, Ar-COCH_3_), 2.78 (t, *J* = 6.5 Hz, 2H, Ar–COCH_2_), 3.33 (t, *J* = 6.5 Hz, 2H, CH_2_), 3.70 (s, 3H, COOCH_3_), 7.69 (d, *J* = 8.0 Hz, 2H, aromatic), 7.70 (d, *J* = 8.5 Hz, 2H, aromatic), 8.03 (d, *J* = 8.0 Hz, 2H, aromatic), 8.06 (d, *J* = 8.5 Hz, 2H, aromatic); **^13^C NMR** (125 MHz, CDCl_3_) δ 26.63 (CH_3_, COCH_3_), 27.99 (CH_2_), 33.45 (CH_2_), 51.81 (CH_3_, O-CH_3_), 127.40 (CH, aromatic), 127.44 (CH, aromatic), 128.70 (CH, aromatic), 128.96 (CH, aromatic), 135.96 (C, aromatic), 136.57 (C, aromatic), 144.26 (C, aromatic), 144.23 (C, aromatic), 173.28 (C, COO), 197.51 (C, CO); analysis for C_19_H_18_O_4_, calculated [M + Na]^+^ (*m*/*z*) = 333.1103 (100.0%), 334.1131 (20.5%), ESI-Q-TOF HR-ESI-MS found: [M + Na]^+^ = 333.1104 (100.0%), 334.1142 (22.46%), δ [ppm] = 0.4.

methyl 4-(4′-hydroxy-[1,1′-biphenyl]-4-yl)-4-oxobutanoate **5l**



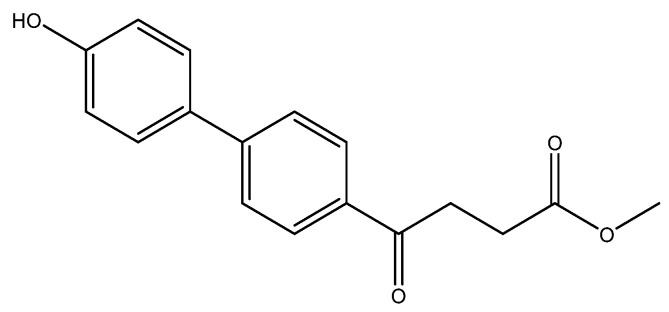



Reagents of **4** (200 mg, 0.85 mmol, 1 eq), 4-bromophenol (294 mg, 1.70 mmol, 2 eq), K_2_CO_3_ (352 mg, 2.55 mmol, 3 eq) and PdCl_2_ (55 mg, 0.31 mmol, 6 mol%) and cosolvents of toluene/EtOH = 1:1 (6 mL) were used. Following the procedure for that of **5d** for 5 h. TLC (acetone/*n*-hexane = 3/7) indicated the consumption of starting material (R*_f_* = 0.14) and formation of the product (R*_f_* = 0.24). The mixture was chromatographed using eluents of CH_3_OH/CH_2_Cl_2_ in a gradient mode of 1/99 → 1/49 → 1/19 to provide the white solid **5l** in 90% yield (214 mg, 0.75 mmol). m.p.: 179–182 °C.

**^1^H NMR** (500 MHz, CDCl_3_/CD_3_OD = 1:1) δ 2.72 (t, *J* = 6.5 Hz, 2H, CH_2_), 3.32 (t, *J* = 6.5 Hz, 2H, CH_2_), 3.67 (s, 3H,-OCH_3_), 6.87 (d, *J* = 8.5 Hz, 2H, aromatic), 7.47 (d, *J* = 8.5 Hz, 2H, aromatic), 7.61 (d, *J* = 8.5 Hz, 2H, aromatic), 7.97 (d, *J* = 8.5 Hz, 2H, aromatic); **^13^C NMR** (125 MHz, CDCl_3_/CD_3_OD = 1:1) δ 28.61 (CH_2_), 33.86 (CH_2_), 52.18 (CH_3_, O-CH_3_), 116.40 (CH, aromatic), 126.99 (CH, aromatic), 128.92 (CH, aromatic), 129.20 (CH, aromatic), 131.61 (C, aromatic), 134.90 (C, aromatic), 146.70 (C, aromatic), 158.26 (C-OH, aromatic), 174.62 (C, COO), 199.26 (C, CO); analysis for C_17_H_16_O_4_, calculated [M + Na]^+^ (*m*/*z*) = 307.0946 (100.0%), 308.0974 (18.4%), 309.1008 (1.6%), [2M + Na]^+^ (*m*/*z*) = 591.1989 (100.0%), 592.2023 (36.8%), 593.2056 (6.6%), ESI-Q-TOF HR-ESI-MS found: [M + Na]^+^ = 307.0947 (94.6%), 308.0987 (31.3%), 309.1035 (2.2%), δ [ppm] = 0.1, [2M + Na]^+^ (*m*/*z*) = 591.1971 (100.0%), 592.2017 (65.4%), 593.2035 (23.8%), δ [ppm] = −3.0.

methyl 4-(2′,4′-dihydroxy-[1,1′-biphenyl]-4-yl)-4-oxobutanoate **5m**



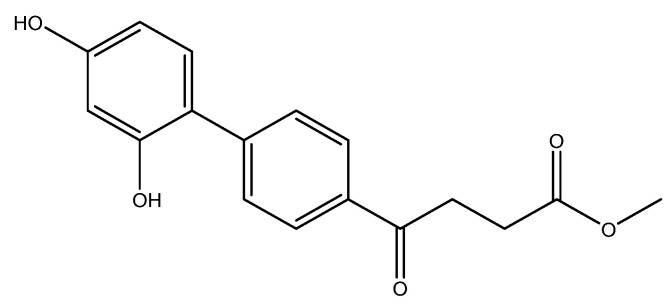



Reagents of **4** (400 mg, 1.69 mmol, 1 eq), 4-bromoresorcinol (642 mg, 3.40 mmol, 2 eq), K_2_CO_3_ (705 mg, 5.10 mmol, 3 eq) and PdCl_2_ (110 mg, 0.62 mmol, 6 mol%) and cosolvents of toluene/EtOH = 1:1 (4.5 mL) were used. Following the procedure for that of **5d** for 19 h. TLC (CH_3_OH/CH_2_Cl_2_ = 1/19) indicated the consumption of starting material (R*_f_* = 0.66) and formation of the product (R*_f_* = 0.50). The mixture was chromatographed using eluents of CH_3_OH/CH_2_Cl_2_ = 1/29 to provide the impure brown solid **5m** in 63% crude yield (321 mg, 1.07 mmol). A portion (110 mg) was purified using HPLC with eluents of CH_3_OH/CH_2_Cl_2_ = 1/49 to afford a white solid of **5m** in 31% yield (55 mg). m.p.: 82–86 °C.

**^1^H NMR** (500 MHz, CDCl_3_/CD_3_OD = 1:1) δ 2.72 (t, *J* = 6.5 Hz, 2H, CH_2_), 3.33 (t, *J* = 6.5 Hz, 2H, CH_2_),3.67 (s, 3H, -OCH_3_), 6.40 (d, *J* = 7.5 Hz, 2H, aromatic), 7.12 (d, *J* = 8.5, 1H, aromatic), 7.65 (d, *J* = 8.5 Hz, 2H, aromatic), 7.94 (d, *J* = 8.5 Hz, 2H, aromatic); **^13^C NMR** (125 MHz, CDCl_3_/CD_3_OD = 1:1) δ 28.70 (CH_2_), 33.91 (CH_2_), 52.17 (O-CH_3_), 103.75 (CH, aromatic), 108.14 (CH, aromatic), 119.84 (C, aromatic), 128.45 (CH, aromatic), 129.85 (CH, aromatic), 131.76 (CH, aromatic), 134.52 (C, aromatic), 145.45 (C, aromatic), 156.11 (C-OH, aromatic), 158.97 (C-OH, aromatic), 174.76 (C, COO), 199.62 (C, CO); analysis for C_17_H_16_O_5_, calculated [M + H]^+^ (*m*/*z*) = 301.1071 (100.0%), 302.1104 (18.4%), [M + Na]^+^ (*m*/*z*) = 323.0890 (100.0%), 324.0923 (18.4%), 325.0957 (1.6%), ESI-Q-TOF HR-ESI-MS found: [M + H]^+^ = 301.1060 (11.3%), 302.1004 (4.0%), δ [ppm] = 3.6, [M + Na]^+^ = 323.0894 (100.0%), 324.0884 (18.7%), 325.1078 (3.6%), δ [ppm] = 1.2.

methyl 4-(4′-nitro-[1,1′-biphenyl]-4-yl)-4-oxobutanoate **5n**



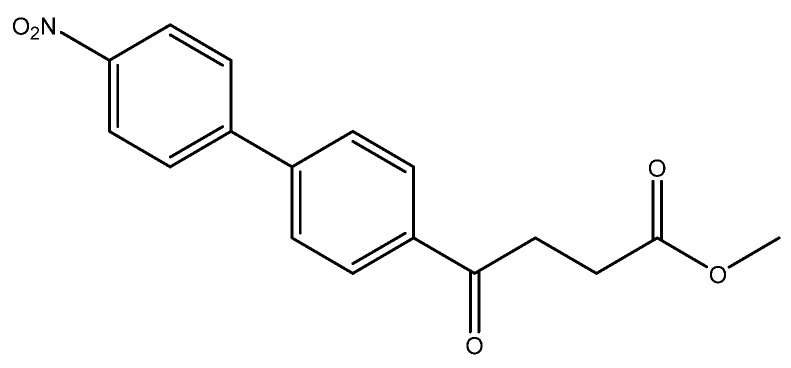



Reagents of **4** (200 mg, 0.85 mmol, 1 eq), bromo-4-nitrobenzene (343.4 mg, 1.70 mmol, 2 eq), K_2_CO_3_ (352 mg, 2.55 mmol, 3 eq) and PdCl_2_ (55 mg, 0.31 mmol, 6 mol%) and cosolvents of toluene/EtOH = 1:1 (7.5 mL) were used. Following the procedure for that of **5d** for 30 min. TLC (acetone/*n*-hexane = 3/7) indicated the consumption of starting material (R*_f_* = 0.20) and formation of the product (R*_f_* = 0.50). The white solid of **5n** was obtained in 93% yield (248 mg, 0.79 mmol). m.p.: 146–150 °C.

**^1^H NMR** (500 MHz, CDCl_3_) δ 2.79 (t, *J* = 6.5 Hz, 2H, -CH_2_), 3.34 (t, *J* = 6.5 Hz, 2H, -CH_2_), 3.70 (s, 3H,-OCH_3_), 7.70 (d, *J* = 8.5 Hz, 2H, aromatic), 7.75 (d, *J* = 8.5 Hz, 2H, aromatic), 8.08 (d, *J* = 8.5 Hz, 2H, aromatic), 8.30 (d, *J* = 8.5 Hz, 2H, aromatic); **^13^C NMR** (125 MHz, CDCl_3_) δ 27.97 (CH_2_), 33.51 (CH_2_), 51.86 (CH_3_, O-CH_3_), 124.20 (CH, aromatic), 127.64 (CH, aromatic), 128.07 (CH, aromatic), 128.85 (CH, aromatic), 136.52 (C, aromatic), 143.22 (C, aromatic), 146.16 (C, aromatic), 147.65 (C-NO_2_, aromatic), 173.26 (C, COO), 197.43 (C, CO); analysis for C_17_H_15_NO_5_, calculated [M + Na]^+^ (*m*/*z*) = 336.0848 (100.0%), 337.0876 (18.4%), ESI-Q-TOF HR-ESI-MS found: [M + Na]^+^ = 336.0846 (100.0%), 337.0880 (20.7%), δ [ppm] = −0.5.

methyl 4-(4′-amino-[1,1′-biphenyl]-4-yl)-4-oxobutanoate **5o**



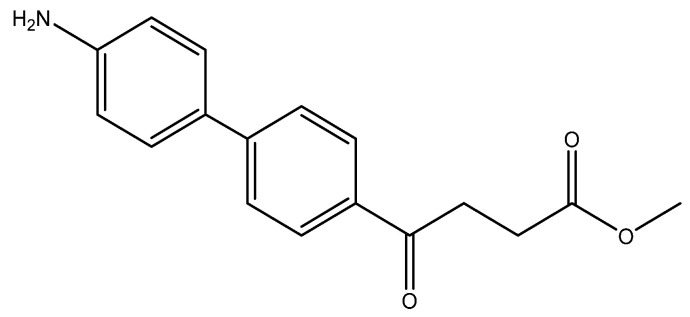



Reagents of **4** (400 mg, 1.69 mmol, 1 eq), 4-bromoaniline (581 mg, 3.38 mmol, 2 eq), K_2_CO_3_ (796 mg, 5.04 mmol, 3 eq) and PdCl_2_ (110 mg, 0.62 mmol, 6 mol%) and cosolvents of toluene/EtOH = 1:1 (6 mL) were used. Following the procedure for that of **5d** for 6 h. TLC (acetone/*n*-hexane = 3/7) indicated the consumption of starting material (R*_f_* = 0.22) and formation of the product (R*_f_* = 0.26). The mixture was chromatographed using eluents of acetone/*n*-hexane/Et_3_N = 2/8/0.3. However, the crude mixture was not dissolved and precipitated in column. After a rough elution of most 4-bromoaniline and a small part of the product, the eluents were changed as EtOAc/Et_3_N = 10/0.3 to dissolve the precipitates. After concentration under reduced pressure, a yellow impure solid was obtained in a quantitative yield (454 mg). m.p.: 94–100 °C.

**^1^H NMR** (500 MHz, CDCl_3_/CD_3_OD = 19:1) δ 2.71 (t, *J* = 6.5 Hz, 2H, CH_2_), 3.27 (t, *J* = 6.5 Hz, 2H, CH_2_), 3.64 (s, 3H,-OCH_3_), 6.86 (d, *J* = 8.5 Hz, 2H, aromatic), 7.44 (d, *J* = 8.5 Hz, 2H, aromatic), 7.56 (d, *J* = 8.5 Hz, 2H, aromatic), 7.94 (d, *J* = 8.5 Hz, 2H, aromatic); **^13^C NMR** (125 MHz, CDCl_3_/CD_3_OD = 19:1) δ 27.96 (CH_2_), 33.21 (CH_2_), 51.74 (CH_3_, O-CH_3_), 116.99 (CH, aromatic), 126.25 (CH, aromatic), 128.16 (CH, aromatic), 128.55 (CH, aromatic), 134.28 (C, aromatic), 145.52 (C-NH_2_, aromatic), 173.63 (C, COO), 197.95 (C, CO); analysis for C_17_H_17_NO_3_, calculated [M + H]^+^ (*m*/*z*) = 284.1287 (100.0%), 285.1315 (18.4%), [M + Na]^+^ (*m*/*z*) = 306.1106 (100.0%), 307.1134 (18.4%), [2M + Na]^+^ (*m*/*z*) = 589.2309 (100.0%), 590.2343 (36.8%), 591.2376 (3.9%), ESI-Q-TOF HR-ESI-MS found: [M + H]^+^ (*m*/*z*) = 284.1292 (51.6%), 285.1350, (14.9%) δ [ppm] = 2.0, [M + Na]^+^ = 306.1106 (66.8%), 307.1165 (17.2%), δ [ppm] = −0.2, [2M + Na]^+^ (*m*/*z*) = 589.2288 (100.0%), 590.2331 (40.9%), 591.2340 (11.4%), δ [ppm] = −3.6.

4-(4′-fluoro-[1,1′-biphenyl]-4-yl)-4-oxobutanoic acid **6a [29]**



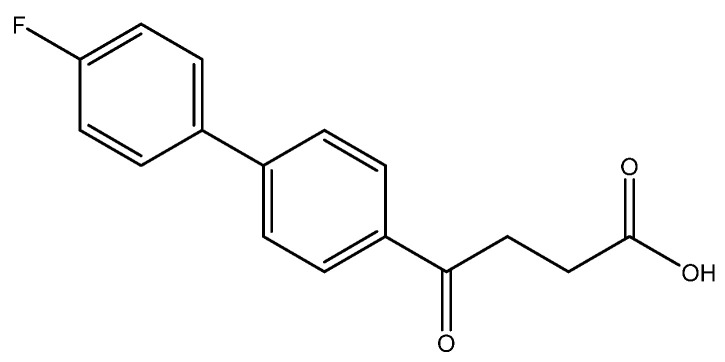



To a two-neck round bottomed flask was added **5a** (15 mg, 0.052 mmol), TFA (1.5 mL) and H_2_O (0.5 mL), sequentially. It was then stirred at 110 °C for 3 h. TLC (CH_3_OH/CH_2_Cl_2_ = 1/19) indicated the consumption of **5a** (R*_f_* = 0.94) and formation of the product **6a** (R*_f_* = 0.34). After concentration under reduced pressure, the residue was chromatographed using eluents of CH_3_OH/CH_2_Cl_2_ in a gradient mode of 1/49 → 1/19 to give a white solid **6a** in 63% yield (9 mg, 0.033 mmol). m.p.: 133–135 °C.

**^1^H NMR** (500 MHz, CDCl_3_/CD_3_OD = 10:1) δ 2.70 (t, *J* = 6.5 Hz, 2H, CH_2_), 3.27 (t, *J* = 6.5 Hz, 2H, CH_2_), 7.09 (t, *J*_2′,3′_ = 8.5 Hz, 2H, aromatic, H-2′), 7.52 (t, *J*_2′,3′_ = 8.5, *J*_HF_ = 3.5 Hz, 2H, aromatic, H-3′), 7.57 (d, *J* = 8.5 Hz, 2H, aromatic), 7.97 (d, *J* = 8.5 Hz, 2H, aromatic); **^13^C NMR** (125 MHz, CDCl_3_/ CD_3_OD = 10:1) δ 27.97 (CH_2_), 33.42 (CH_2_), 115.80 (d, ^2^*J*_CF_ = 21.5 Hz, CH, aromatic), 127.02 (CH, aromatic), 128.61 (CH, aromatic), 128.84 (d, ^3^*J*_CF_ = 8.1 Hz, CH, aromatic), 135.16 (C, aromatic), 135.87 (d, ^4^*J*_CF_ = 2.6 Hz, C, aromatic), 144.88 (C, aromatic), 162.95 (d, ^1^*J*_CF_ = 246.5 Hz, C, aromatic) 175.21 (C, COO), 198.35 (C, CO); analysis for C_16_H_13_FO_3_, calculated [M − H]^−^ (*m*/*z*) = 271.0771 (100.0%), 272.0810 (17.3%), 273.0843 (1.4%), ESI-Q-TOF HR-ESI-MS found: [M − H]^−^ = 271.0770 (100.0%), 272.0805 (17.3%), 273.0833 (2.1%), δ [ppm] = −0.15.

4-(4′-fluoro-2′-methyl-[1,1′-biphenyl]-4-yl)-4-oxobutanoic acid **6b**



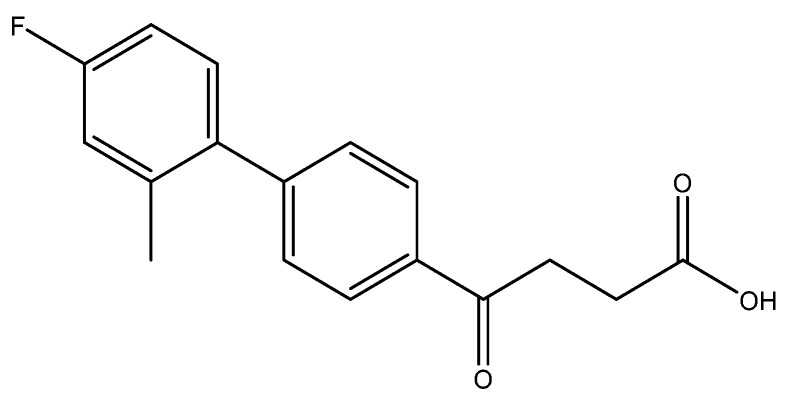



Reagents of **5b** (45 mg, 0.15 mmol), TFA (1.5 mL) and H_2_O (1.5 mL) were used. The procedure followed that described for preparing **6a**. Reaction was allowed for 4 h and TLC (CH_3_OH/CH_2_Cl_2_ = 1/19) indicated the consumption of the starting material (R*_f_* = 0.94) and formation of the product **6b** (R*_f_* = 0.42). After column chromatography, a white solid **6b** was obtained in 73% yield (30 mg, 0.11 mmol). 135–138 °C.

**^1^H NMR** (500 MHz, CD_3_OD/CDCl_3_ = 1:1) δ 2.24 (s, 3H, Ar-CH_3_), 2.72 (t, *J* = 6.5 Hz, 2H, CH_2_), 3.49 (t, *J* = 6.5 Hz, 2H, CH_2_), 6.97 (td, *J* = 8.5, *J*_HF_ = 2.5 Hz, 1H, aromatic), 7.04 (dd, *J* = 9.5, *J* = 2.5 Hz, 1H, aromatic), 7.21 (dd, *J* = 8.0, *J*_HF_ = 6.0 Hz, 1H, aromatic), 7.42 (d, *J* = 8.0 Hz, 2H, aromatic), 8.06 (d, *J* = 8.0 Hz, 2H, aromatic); **^13^C NMR** (125 MHz, CDCl_3_/CD_3_OD = 1:1) δ 20.60 (Ar-CH_3_), 29.05 (CH_2_), 34.52 (CH_2_), 51.84 (CH_3_, O-CH_3_), 113.65 (d, ^1^*J*_CF_ = 21.4 Hz, CH, aromatic), 117.85 (d, ^2^*J*_CF_ = 21.4 Hz, CH, aromatic), 129.12 (CH, aromatic), 130.70 (CH, aromatic), 132.20 (d, ^3^*J*_CF_ = 8.4 Hz, CH, aromatic), 136.75 (C, aromatic), 138.50 (d, ^4^*J*_CF_ = 2.5 Hz, C, aromatic), 139.09 (d, ^3^*J*_CF_ = 7.9 Hz, C, aromatic), 147.44 (C, aromatic), 163.76 (d, ^1^*J*_CF_ = 243.6 Hz, C, aromatic) 176.72 (C, COOH), 200.19 (C, Ar-CO); analysis for C_17_H_15_FO_3_, calculated [M − H]^−^ (*m*/*z*) = 285.0927 (100.0%), 286.0966 (18.4%), 287.1000 (1.6%), ESI-Q-TOF HR-ESI-MS found: [M − H]^−^ = 285.0922 (100.0%), 286.0957 (18.1%), 287.0978 (2.1%), δ [ppm] = −1.7.

4-(2′,4′-difluoro-[1,1′-biphenyl]-4-yl)-4-oxobutanoic acid **6c**



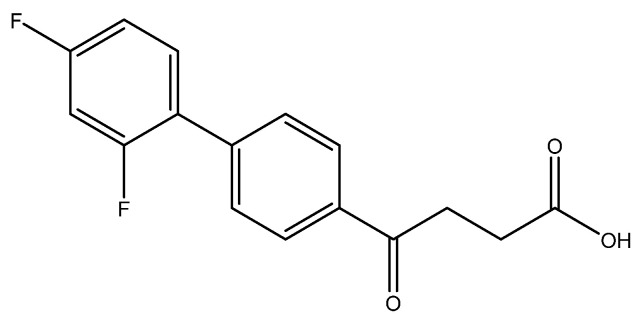



Reagents of **5c** (96 mg, 0.32 mmol), TFA (1 mL) and H_2_O (1 mL) were used. The procedure followed that described for preparing **6a**. Reaction was allowed for 4 h and TLC (CH_3_OH/CH_2_Cl_2_ = 1/19) indicated the consumption of the starting material (R*_f_* = 0.98) and formation of the product **6c** (R*_f_* = 0.32). After column chromatography, a white solid **6c** was obtained in 93% yield (86 mg, 0.30 mmol). 112–115 °C.

**^1^H NMR** (500 MHz, CDCl_3_/CD_3_OD = 1:1) δ 2.74 (t, *J* = 6.5 Hz, 2H, CH_2_), 3.35 (t, *J* = 6.5 Hz, 2H, CH_2_), 6.97 (m, 2H, aromatic), 7.46 (q, *J* = 8.5 Hz, 1H, aromatic), 7.59 (m, 2H, aromatic), 8.02 (d, *J* = 8.5 Hz, 2H, aromatic); **^13^C NMR** (125 MHz, CDCl_3_/CD_3_OD = 1:1) δ 27.23 (CH_2_), 32.69 (CH_2_), 51.83 (CH_3_, O-CH_3_), 103.69 (t, ^2^*J*_CF_ = 26.0 Hz, CH, C-3′), 111.14 (dd, ^2^*J*_CF_ = 21.3 Hz, ^4^*J*_CF_ = 3.5 Hz, CH, C-5′), 123.52 (dd, ^2^*J*_CF_ = 13.1 Hz, ^4^*J*_CF_ = 3.5 Hz, C, C-1′) 127.56 (CH, aromatic), 128.44 (CH, aromatic), 130.93 (dd, ^3^*J*_CF_ = 9.5 Hz, ^3^*J*_CF_ = 4.4 Hz, CH, C-6′), 134.97 (C, aromatic), 139.35 (C, aromatic), 159.45 (dd, ^1^*J*_CF_ = 250.0, ^3^*J*_CF_ = 11.9 Hz, C-F, C-2′), 162.32 (dd, ^1^*J*_CF_ = 248.4, ^3^*J*_CF_ = 11.9 Hz, C-F, C-4′), 173.33 (C, COO), 197.96 (C, CO); analysis for C_16_H_11_F_2_O_3_, calculated [M − H]^−^ (*m*/*z*) = 289.0682 (100.0%), 290.0715 (17.3%), 291.0749 (1.4%), ESI-Q-TOF HR-ESI-MS found: [M − H]^−^ = 289.0662 (100.0%), 290.0697 (17.1%), 291.0716 (2.0%), δ [ppm] = −6.9.

4-(4′-chloro-[1,1′-biphenyl]-4-yl)-4-oxobutanoic acid **6d [49]**



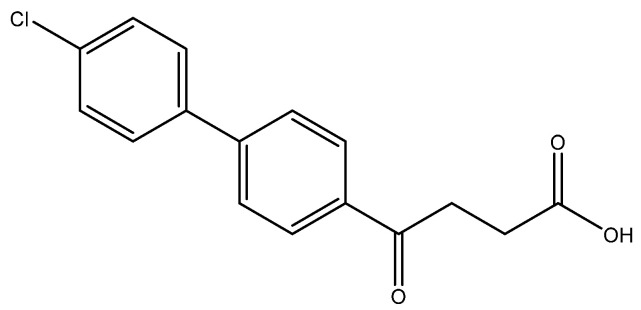



Reagents of **5d** (100 mg, 0.33 mmol), TFA (1 mL) and H_2_O (1 mL) were used. The procedure followed that described for preparing **6a**. Reaction was allowed for 4 h and TLC (CH_3_OH/CH_2_Cl_2_ = 1:19) indicated the consumption of the starting material (R*_f_* = 0.98) and formation of the product **6c** (R*_f_* = 0.32). After column chromatography, a white solid **6d** was obtained in 85% yield (80 mg, 0.28 mmol). m.p.: 160–163 °C.

**^1^H NMR** (500 MHz, CDCl_3_/CD_3_OD = 1:1) δ 2.72 (t, *J* = 6.5 Hz, 2H, CH_2_), 3.32 (t, *J* = 6.5 Hz, 2H, CH_2_), 7.40 (dd, *J* = 8.5, 2.0 Hz, 2H, aromatic), 7.55 (dd, *J* = 8.5, 2.0 Hz, 2H, aromatic), 7.65 (d, *J* = 8.5 Hz, 2H, aromatic), 8.02 (d, *J* = 8.5 Hz, 2H, aromatic); **^13^C NMR** (125 MHz, CDCl_3_/CD_3_OD = 1:1) δ 28.56 (CH_2_), 34.06 (CH_2_), 127.64 (CH, aromatic), 129.08 (CH, aromatic), 129.29 (CH, aromatic), 129.64 (CH, aromatic), 135.02 (C-Cl, aromatic), 136.17 (C, aromatic), 138.91 (C, aromatic), 145.29 (C, aromatic), 175.87 (C, COOH), 199.37 (C, Ar-CO); analysis for C_16_H_13_ClO_3_, calculated [M − H]^−^ (*m*/*z*) = 287.0475 (100.0%), 289.0451 (32.0%), 288.0514 (17.3%), 290.0485 (5.5%) ESI-Q-TOF HR-ESI-MS found: [M − H]^−^ = 287.0475 (51.7%), 289.0459 (22.32%), 288.0507 (16.3%), 290.0474 (1.9%) δ [ppm] = 0.07.

4-(4′-chloro-2′-methyl-[1,1′-biphenyl]-4-yl)-4-oxobutanoic acid **6e**



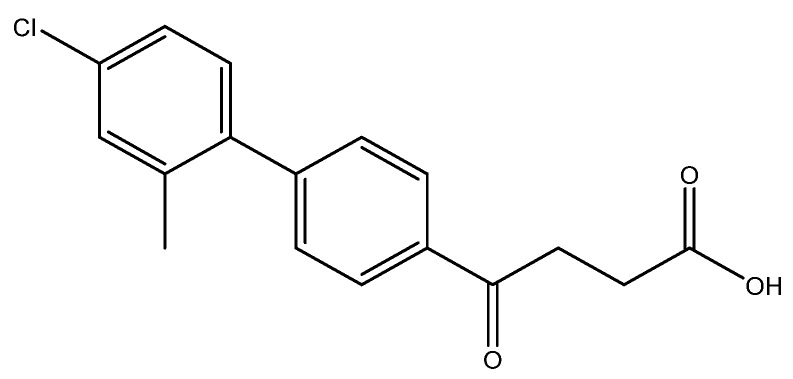



Reagents of **5e** (30 mg, 0.095 mmol), TFA (1.5 mL) and H_2_O (0.5 mL) were used. The procedure followed that described for preparing **6a**. Reaction was allowed for 1.5 h and TLC (CH_3_OH/CH_2_Cl_2_ = 1/19) indicated the consumption of the starting material (R*_f_* = 0.98) and formation of the product **6e** (R*_f_* = 0.46). After column chromatography, a white solid **6e** was obtained in 77% yield (22 mg, 0.073 mmol).

**^1^H NMR** (500 MHz, CDCl_3_/CD_3_OD = 1:1) δ 2.21 (s, 3H, Ar-CH_3_), 2.73 (t, *J* = 6.5 Hz, 2H, CH_2_), 3.33 (t, *J* = 6.5 Hz, 2H, CH_2_), 7.12 (d, *J* = 8.5 Hz, 1H, aromatic), 7.20 (dd, *J* = 8.5, 2.0 Hz, 1H, aromatic), 7.25 (d, *J* = 2.0 Hz, 1H, aromatic), 7.38 (dd, *J* = 8.5, 2.0 Hz, 2H, aromatic), 8.02 (d, *J* = 8.5 Hz, 2H, aromatic); **^13^C NMR** (125 MHz, CDCl_3_/CD_3_OD = 1:1) δ 20.47 (Ar-CH_3_), 28.62 (CH_2_), 34.05 (CH_2_), 126.63 (CH, aromatic), 128.72 (CH, aromatic), 130.13 (CH, aromatic), 130.89 (CH, aromatic), 131.35 (CH, aromatic), 134.27 (C, aromatic), 135.97 (C-Cl, aromatic), 137.86 (C, aromatic), 139.98 (C, aromatic), 146.81 (C, aromatic), 174.68 (C, COO), 199.40 (C, CO); analysis for C_17_H_15_ClO_3_, calculated [M − H]^−^ (*m*/*z*) = 301.0632 (100.0%), 303.0607 (32.0%), 302.0671 (18.4%), 304.0641 (5.9%), ESI-Q-TOF HR-ESI-MS found: [M − H]^−^ = 301.0631 (100.0%), 303.0609 (33.7%), 302.0669 (19.7%), 304.0640 (7.2%), δ [ppm] = −0.06.

4-(2′,4′-dichloro-[1,1′-biphenyl]-4-yl)-4-oxobutanoic acid **6f**



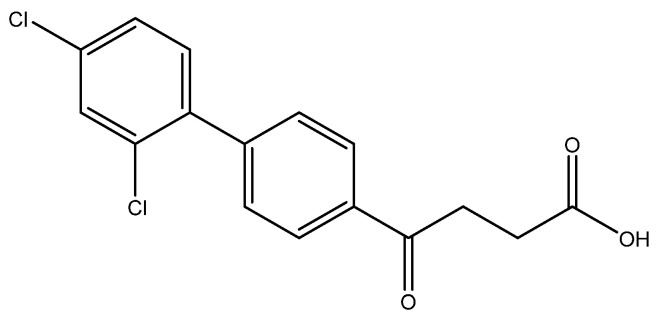



Reagents of **5f** (105 mg, 0.31 mmol), TFA (1 mL) and H_2_O (1 mL) were used. The procedure followed that described for preparing **6a**. Reaction was allowed for 4 h and TLC (CH_3_OH/CH_2_Cl_2_ = 1/19) indicated the consumption of the starting material (R*_f_* = 0.92) and formation of the product **6f** (R*_f_* = 0.36). After column chromatography, a white solid **6f** was obtained in 84% yield (84 mg, 0.26 mmol). m.p.: 135–137 °C.

**^1^H NMR** (500 MHz, CDCl_3_/CD_3_OD = 1:1) δ 2.75 (t, *J* = 6.5 Hz, 2H, CH_2_), 3.36 (t, *J* = 6.5 Hz, 2H, CH_2_), 7.29 (d, *J* = 8.0, Hz, 1H, aromatic), 7.33 (dd, *J* = 8.0, 2.0 Hz, 1H, aromatic), 7.50 (dt, *J* = 7.0, 2.0 Hz, 3H, aromatic), 8.02 (d, *J* = 8.0 Hz, 2H, aromatic); **^13^C NMR** (125 MHz, CDCl_3_/CD_3_OD = 1:1) δ 27.22 (CH_2_), 32.73 (CH_2_), 126.72 (CH, aromatic), 127.24 (CH, aromatic), 129.02 (CH, aromatic), 129.05 (CH, aromatic), 131.31 (CH, aromatic), 132.33 (C-Cl, aromatic), 133.82 (C-Cl, aromatic), 135.15 (C, aromatic), 137.37 (C, aromatic), 142.67 (C, aromatic), 173.31 (C, COO), 197.96 (C, CO); analysis for C_16_H_12_Cl_2_O_3_, calculated [M − H]^−^ (*m*/*z*) = 321.0085 (100.0%), 323.0061 (63.9%), 322.0124 (17.3%), 324.0095 (11.1%), 325.0032 (10.2%), 326.0065 (1.8%), ESI-Q-TOF HR-ESI-MS found: [M − H]^−^ = 321.0069 (100.0%), 323.0041 (60.2%), 322.0102 (17.3%), 324.0072 (10.7%), 325.0017 (10.6%), 326.0036 (2.3%), δ [ppm] = −1.3.

4-(3′,4′-dibromo-[1,1′-biphenyl]-4-yl)-4-oxobutanoic acid **6g**



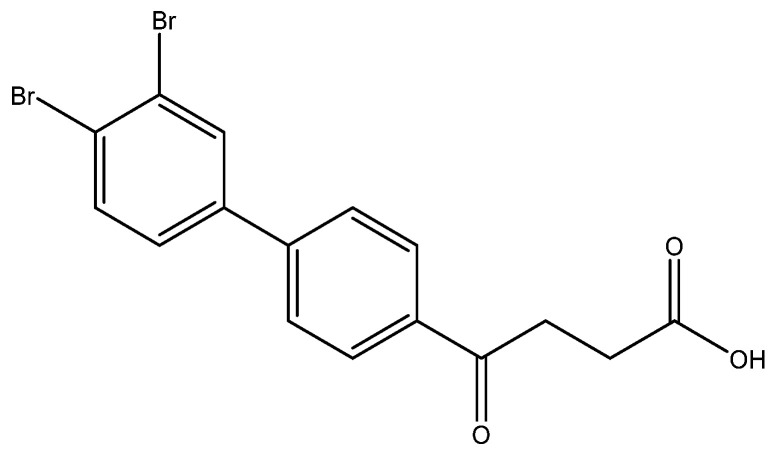



Reagents of **5g** (62 mg, 0.15 mmol), TFA (1.5 mL) and H_2_O (0.5 mL) were used. The procedure followed that described for preparing **6a**. Reaction was allowed for 2 h and TLC (CH_3_OH/CH_2_Cl_2_ = 1/19) indicated the consumption of the starting material (R*_f_* = 0.98) and formation of the product **6e** (R*_f_* = 0.40). After column chromatography, a white solid **6g** was obtained in 80% yield (50 mg, 0.12 mmol). 137–140 °C.

**^1^H NMR** (500 MHz, CDCl_3_/CD_3_OD) δ 2.72 (t, *J* = 6.5 Hz, 2H, CH_2_), 3.31 (t, *J* = 6.5 Hz, 2H, CH_2_), 7.43 (d, *J* = 8.5 Hz, *J* = 2.0 Hz, 1H, aromatic), 7.64 (d, *J* = 8.5 Hz, 2H, aromatic), 7.68 (d, *J* = 8.5 Hz, 2H, aromatic), 7.86 (s, *J* = 2.0 Hz, 1H, aromatic), 8.03 (d, *J* = 8.5 Hz, 2H, aromatic); **^13^C NMR** (125 MHz, CDCl_3_) δ 28.61 (CH_2_), 34.15 (CH_2_), 125.22 (C-Br, aromatic), 125.89 (C-Br, aromatic), 127.71 (CH, aromatic), 127.94 (CH, aromatic), 129.40 (CH, aromatic), 132.76 (CH, aromatic), 134.74 (CH, aromatic), 136.68 (C, aromatic), 141.27 (C, aromatic), 143.88 (C, aromatic), 175.97 (C, COO), 199.34 (C, CO); analysis for C_16_H_12_Br_2_O_3_, calculated [M − H]^−^ (*m*/*z*) = 410.9060 (100.0%), 408.9080 (51.4%), 412.9039 (48.6%), 411.9094 (9.7%), 413.9073 (8.4%), ESI-Q-TOF HR-ESI-MS found: [M − H]^−^ = 410.9046 (100.0%), 408.9067 (50.8%), 412.9029 (50.3%), 411.9080 (17.4%), 413.9062 (10.1%), δ [ppm] = −3.4.

4-(2′,5′-dibromo-[1,1′-biphenyl]-4-yl)-4-oxobutanoic acid **6h**



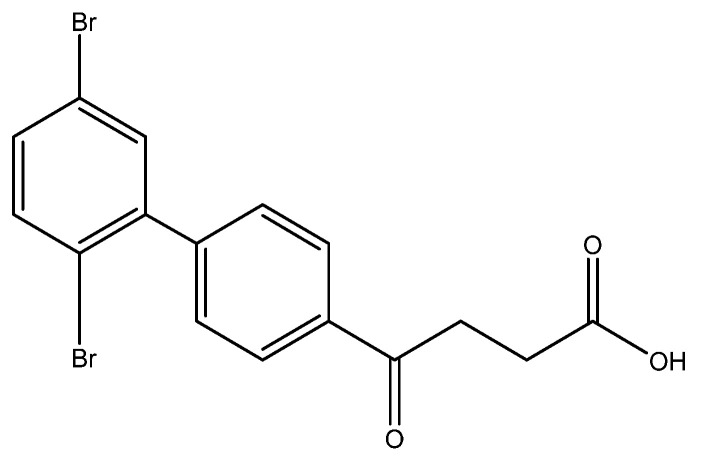



Reagents of **5h** (25 mg, 0.06 mmol), TFA (1.0 mL) and H_2_O (1.0 mL) were used. The procedure followed that described for preparing **6a**. Reaction was allowed for 4 h and TLC (CH_3_OH/CH_2_Cl_2_ = 1/19) indicated the consumption of the starting material (R*_f_* = 0.98) and formation of the product **6h** (R*_f_* = 0.42). After column chromatography, a white solid **6h** was obtained in 67% yield (17 mg, 0.04 mmol). m.p.: 77–80 °C.

**^1^H NMR** (500 MHz, CDCl_3_/CD_3_OD = 2:1) δ 2.73 (t, *J* = 6.5 Hz, 2H, CH_2_), 3.33 (t, *J* = 6.5 Hz, 2H, CH_2_), 3.70 (s, 3H,-OCH_3_), 7.19 (d, *J* = 8.0 Hz, 1H, aromatic), 7.47 (dd, *J* = 8.5, 2.0 Hz, 2H, aromatic) 7.51 (dd, *J* = 8.0, 2.0 Hz, 1H, aromatic), 7.82 (d, *J* = 2.0 Hz, 1H, aromatic), 8.02 (d, *J* = 8.5 Hz, 2H, aromatic); **^13^C NMR** (125 MHz, CDCl_3_/CD_3_OD = 2:1) δ 27.46 (CH_2_), 32.95 (CH_2_), 121.75 (C-Br, aromatic), 122.24 (C-Br, aromatic), 127.39 (CH, aromatic), 129.12 (CH, aromatic), 130.29 (CH, aromatic), 131.53 (CH, aromatic), 134.99 (CH, aromatic), 135.36 (C, aromatic), 139.98 (C, aromatic), 144.46 (C, aromatic), 174.79 (C, COO), 198.32 (C, CO); analysis for C_16_H_12_Br_2_O_3_, calculated [M − H]^−^ (*m*/*z*) = 410.9060 (100.0%), 408.9080 (51.4%), 412.9039 (48.6%), 411.9094 (9.7%), 413.9073 (8.4%), ESI-Q-TOF HR-ESI-MS found: [M − H]^−^ = 410.9043 (100.0%), 408.9065 (49.8%), 412.9030 (52.0%), 411.9087 (25.8%), 413.9068 (14.5%), δ [ppm] = −4.1.

4-(2′,4′-dibromo-[1,1′-biphenyl]-4-yl)-4-oxobutanoic acid **6i**



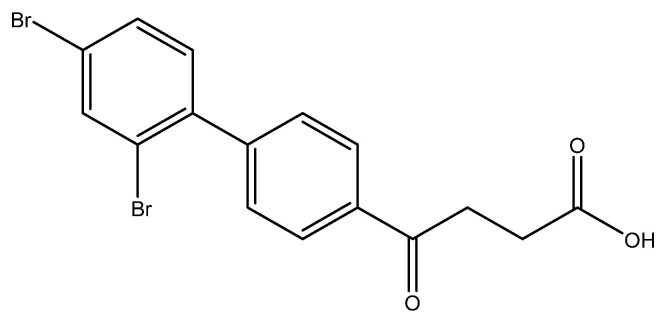



Reagents of **5i** (45 mg, 0.11 mmol), TFA (1.5 mL) and H_2_O (0.5 mL) were used. The procedure followed that described for preparing **6a**. Reaction was allowed for 4 h and TLC (CH_3_OH/CH_2_Cl_2_ = 1/19) indicated the consumption of the starting material (R*_f_* = 0.98) and formation of the product **6i** (R*_f_* = 0.32). After column chromatography, a white solid **6i** was obtained in 69% yield (31 mg, 0.076 mmol). m.p.: 108–110 °C.

**^1^H NMR** (500 MHz, CDCl_3_) δ 2.74 (t, *J* = 6.5 Hz, 2H, CH_2_), 3.34 (t, *J* = 6.5 Hz, 2H, CH_2_), 7.37 (dd, *J* = 8.5 Hz, *J* = 2.0 Hz, 1H, aromatic), 7.44 (d, *J* = 2.0 Hz, 1H, aromatic), 7.48 (d, *J* = 8.5 Hz, 2H, aromatic), 7.54 (d, *J* = 8.0 Hz, 1H, aromatic), 8.03 (d, *J* = 8.0 Hz, 2H, aromatic); **^13^C NMR** (125 MHz, CDCl_3_) δ 28.62 (CH_2_), 34.21 (CH_2_), 121.52 (C-Br, aromatic), 121.95 (C-Br, aromatic), 128.58 (CH, aromatic), 130.30 (CH, aromatic), 133.06 (CH, aromatic), 134.25 (CH, aromatic), 135.37 (CH, aromatic), 136.82 (CH, aromatic), 144.07 (C, aromatic), 145.37 (C, aromatic), 175.98 (C, COO), 199.51 (C, CO); analysis for C_16_H_12_Br_2_O_3_, calculated [M − H]^−^ (*m*/*z*) = 410.9060 (100.0%), 408.9080 (51.4%), 412.9039 (48.6%), 411.9094 (9.7%), 413.9073 (8.4%), ESI-Q-TOF HR-ESI-MS found: [M − H]^−^ = 410.9062 (100.0%), 408.9071 (43.5%), 412.9036 (55.3%), 411.9074 (13.4%), 413.9054 (9.4%), δ [ppm] = 0.5.

4-(3′,5′-dibromo-[1,1′-biphenyl]-4-yl)-4-oxobutanoic acid **6j**



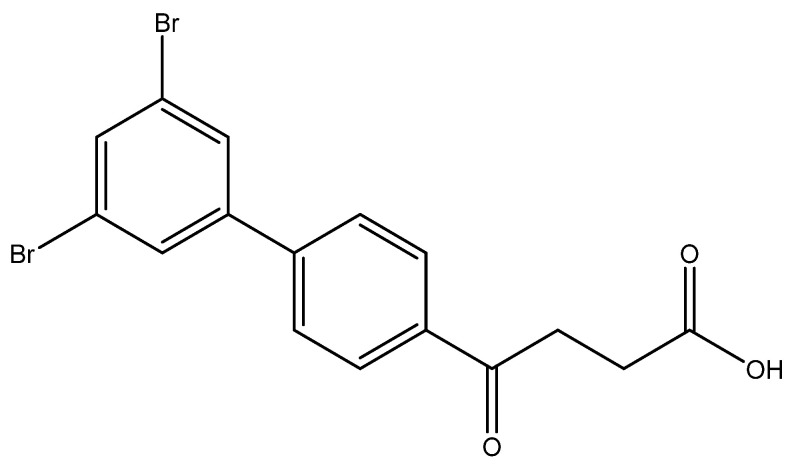



Reagents of **5j** (85 mg, 0.20 mmol), TFA (1.0 mL) and H_2_O (1.0 mL) were used. The procedure followed that described for preparing **6a**. Reaction was allowed for 3 h and TLC (CH_3_OH/CH_2_Cl_2_ = 1/19) indicated the consumption of the starting material (R*_f_* = 0.98) and formation of the product **6j** (R*_f_* = 0.34). After column chromatography, a white solid **6j** was obtained in 60% yield (48 mg, 0.12 mmol). m.p.: 181–183 °C.

**^1^H NMR** (500 MHz, CDCl_3_/CD_3_OD = 1:1) δ 2.74 (t, *J* = 6.5 Hz, 2H, CH_2_), 3.30 (t, *J* = 6.5 Hz, 2H, CH_2_), 7.60 (dd, *J* = 8.5, 2.0 Hz, 2H, aromatic), 7.65 (s, 3H, aromatic), 8.02 (d, *J* = 8.5 Hz, 2H, aromatic); **^13^C NMR** (125 MHz, CDCl_3_/CD_3_OD = 1:1) δ 28.25 (CH_2_), 33.73 (CH_2_), 123.68 (C-Br, aromatic), 127.62 (CH, aromatic), 129.04 (CH, aromatic), 129.32 (CH, aromatic), 133.76 (CH, aromatic), 136.38 (C, aromatic), 143.23 (C, aromatic), 143.63 (C, aromatic), 174.04 (C, COO), 198.44 (C, CO); analysis for C_16_H_12_Br_2_O_3_, calculated [M − H]^−^ (*m*/*z*) = 410.9060 (100.0%), 408.9080 (51.4%), 412.9039 (48.6%), 411.9094 (9.7%), 413.9073 (8.4%), 409.9114 (4.4%), ESI-Q-TOF HR-ESI-MS found: [M − H]^−^ = 410.9021 (100.0%), 408.9042 (50.6%), 412.9003 (46.7%), 411.9054 (16.1%), 413.9029 (12.4%), 409.9078 (8.6%), δ [ppm] = −9.5.

4-(4′-acetyl-[1,1′-biphenyl]-4-yl)-4-oxobutanoic acid **6k**



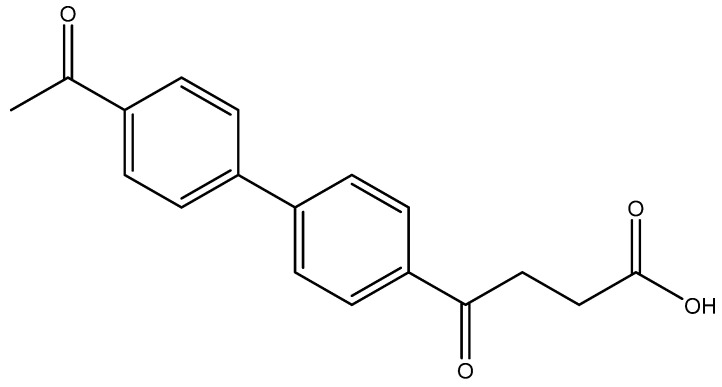



Reagents of **5k** (100 mg, 0.32 mmol), TFA (1.5 mL) and H_2_O (1.5 mL) were used. The procedure followed that described for preparing **6a**. Reaction was allowed for 3 h and TLC (CH_3_OH/CH_2_Cl_2_ = 1/19) indicated the consumption of the starting material (R*_f_* = 0.98) and formation of the product **6j** (R*_f_* = 0.34). After column chromatography, a white solid **6k** was obtained in 88% yield (83 mg, 0.28 mmol). m.p.: 203–205 °C.

**^1^H NMR** (500 MHz, DMSO-d6) δ 2.60 (t, *J* = 6.5 Hz, 5H, Ar–COCH_2_), 3.28 (t, *J* = 6.5 Hz, 2H, CH_2_), 7.89 (d, *J* = 8.0 Hz, 4H, aromatic), 8.05 (d, *J* = 8.5 Hz, 2H, aromatic), 8.08 (d, *J* = 8.5 Hz, 2H, aromatic), 12.14 (s, 1H, -COOH); **^13^C NMR** (125 MHz, DMSO-d6) δ 26.74 (CH_3_, COCH_3_), 27.85 (CH_2_), 33.17 (CH_2_), 127.19 (CH, aromatic), 127.23 (CH, aromatic), 128.57 (CH, aromatic), 128.289 (CH, aromatic), 135.91 (C, aromatic), 136.28 (C, aromatic), 143.13 (C, aromatic), 173.72 (C, COOH), 197.45 (C, CO), 198.03 (C, CO); analysis for C_18_H_16_O_4_, calculated [M − H]^−^ (*m*/*z*) = 295.0970 (100.0%), 296.1009 (19.5%), 297.1043 (1.8%), ESI-Q-TOF HR-ESI-MS found: [M − H]^−^ = 295.0968 (100.0%), 296.1003 (18.3%), 297.1032 (2.6%), δ [ppm] = −0.8.

4-(4′-hydroxy-[1,1′-biphenyl]-4-yl)-4-oxobutanoic acid **6l [25]**



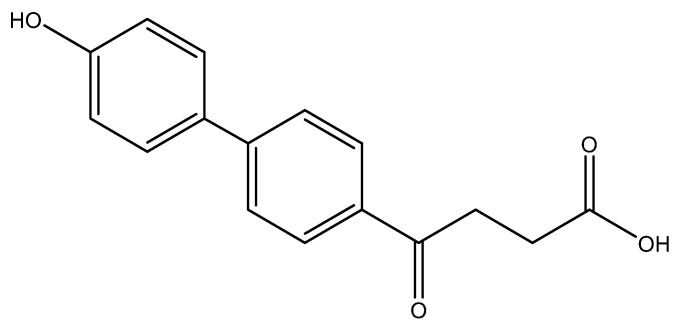



Reagents of **5l** (74 mg, 0.26 mmol), TFA (3.0 mL) and H_2_O (2.0 mL) were used. The procedure followed that described for preparing **6a**. Reaction was allowed for 4 h and TLC (CH_3_OH/CH_2_Cl_2_ = 1/19) indicated the consumption of the starting material (R*_f_* = 0.68) and formation of the product **6l** (R*_f_* = 0.26). After column chromatography, a white solid **6l** was obtained in 81% yield (56 mg, 0.21 mmol). 217–220 °C.

**^1^H NMR** (500 MHz, CDCl_3_/CD_3_OD = 1:1) δ 2.72 (t, *J* = 6.5 Hz, 2H, CH_2_), 3.30 (t, *J* = 6.5 Hz, 2H, CH_2_), 6.87 (d, *J* = 8.5 Hz, 2H, aromatic), 7.46 (d, *J* = 8.5 Hz, 2H, aromatic), 7.61 (d, *J* = 8.5 Hz, 2H, aromatic), 7.96 (d, *J* = 8.5 Hz, 2H, aromatic); **^13^C NMR** (125 MHz, CDCl_3_/CD_3_OD = 1:1) δ 28.73 (CH_2_), 33.94 (CH_2_), 116.49 (CH, aromatic), 127.03 (CH, aromatic), 128.91 (CH, aromatic), 129.19 (CH, aromatic), 131.76 (C, aromatic), 135.14 (C, aromatic), 146.77 (C, aromatic), 158.34 (C-OH, aromatic), 174.59 (C, COO), 199.32 (C, CO); analysis for C_14_H_14_O_4_, calculated [M − H]^−^ (*m*/*z*) = 269.0814 (100.0%), 270.0853 (17.3%), 271.0886 (1.4%), ESI-Q-TOF HR-ESI-MS found: [M − H]^−^ = 269.0810 (100.0%), 270.0843 (17.5%), 271.0868 (2.0%), δ [ppm] = −1.4.

4-(2′,4′-dihydroxy-[1,1′-biphenyl]-4-yl)-4-oxobutanoic acid **6m**



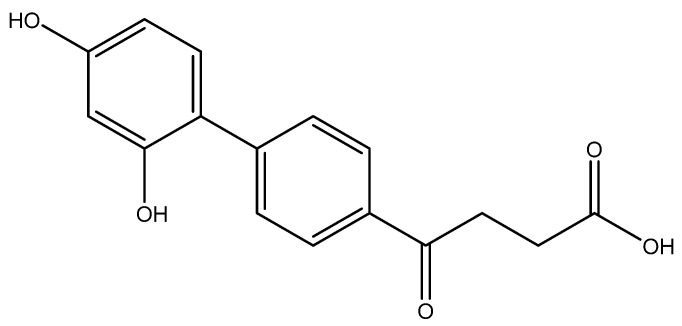



Reagents of **5m** (36 mg, 0.12 mmol), a solution of aqueous THF (1.2 mL, 75% vol) were used. The gold solution was cooled at 0 °C and LiOH (10 mg, 0.42 mmol, 3.5 eq) was added. After 10 min, the ice bath was removed and the reaction was allowed for 30 min and it turned dark green. A further 30 min reaction indicated the consumption of the starting material (R*_f_* = 0.70) and formation of the product **6m** (R*_f_* = 0.38) from TLC (CH_3_OH/CH_2_Cl_2_ = 1/9). It was treated with cationic exchange resin (H^+^) and the solution turned orange. Followed by filtration, concentration of the filtrate and column chromatography (CH_3_OH/CH_2_Cl_2_ = 1/9) of the residue, a white solid **6m** was obtained in 50% yield (18 mg, 0.06 mmol). m.p.: 210–212 °C.

**^1^H NMR** (500 MHz, CD_3_OD) δ 2.70 (t, *J* = 6.5 Hz, 2H, CH_2_), 3.32 (t, *J* = 6.5 Hz, 2H, CH_2_), 6.39 (d, *J* = 8.0 Hz, 2H, aromatic), 7.14 (d, *J* = 8.0 Hz, 1H, aromatic), 7.66 (d, *J* = 8.5 Hz, 2H, aromatic), 7.97 (d, *J* = 8.5 Hz, 2H, aromatic); **^13^C NMR** (125 MHz, CD_3_OD) δ 29.06 (CH_2_), 34.36 (CH_2_), 104.11 (CH, aromatic), 108.51 (CH, aromatic), 120.29 (C, aromatic), 128.78 (CH, aromatic), 130.20 (CH, aromatic), 132.25 (CH, aromatic), 135.32 (C, aromatic), 145.94 (C, aromatic), 156.83 (C-OH, aromatic), 1589.82 (C-OH, aromatic), 176.77 (C, COOH), 200.36 (C, CO); analysis for C_16_H_14_O_5_, calculated [M − H]^−^ (*m*/*z*) = 285.0768 (100.0%), 286.0802 (17.3%), ESI-Q-TOF HR-ESI-MS found: [M − H]^−^ = 285.0764 (100%), 286.0783 (19.5%), δ [ppm] = −1.4.

4-(4′-nitro-[1,1′-biphenyl]-4-yl)-4-oxobutanoic acid **6n**



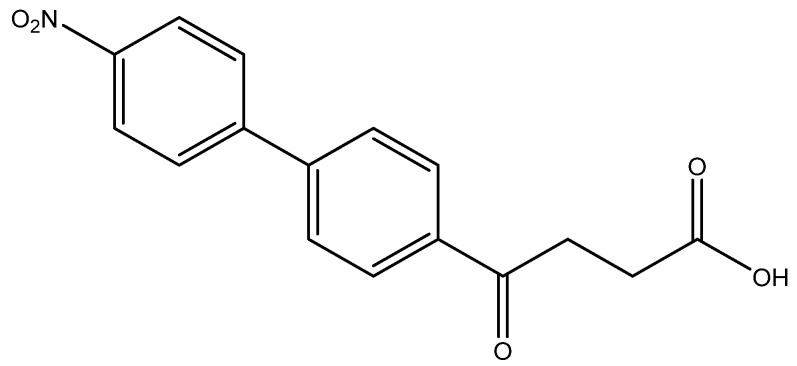



Reagents of **5n** (100 mg, 0.32 mmol), TFA (2.5 mL) and H_2_O (1.0 mL) were used. The procedure followed that described for preparing **6a**. Reaction was allowed for 4 h and TLC (CH_3_OH/CH_2_Cl_2_ = 1/19) indicated the consumption of the starting material (R*_f_* = 0.98) and formation of the product **6n** (R*_f_* = 0.40). After column chromatography, a white solid **6n** was obtained in 72% yield (70 mg, 0.23 mmol). m.p.: 157–160 °C.

**^1^H NMR** (500 MHz, CDCl_3_/CD_3_OD = 1:1) δ 2.73 (t, *J* = 6.5 Hz, 2H, CH_2_), 3.34 (t, *J* = 6.5 Hz, 2H, CH_2_), 7.76 (d, *J* = 8.5 Hz, 2H, aromatic), 7.82 (d, *J* = 9.0 Hz, 2H, aromatic), 8.08 (d, *J* = 8.5 Hz, 2H, aromatic), 8.29 (d, *J* = 8.5 Hz, 2H, aromatic); **^13^C NMR** (125 MHz, CDCl_3_/CD_3_OD = 1:1) δ 28.63 (CH_2_), 34.26 (CH_2_), 124.80 (CH, aromatic), 128.36 (CH, aromatic), 128.86 (CH, aromatic), 129.53 (CH, aromatic), 137.36 (C, aromatic), 144.04 (C, aromatic), 146.96 (C, aromatic), 148.45 (C-NO_2_, aromatic), 176.01 (C, COO), 199.44 (C, CO); analysis for C_16_H_13_NO_5_, calculated [M − H]^−^ (*m*/*z*) = 298.0716 (100.0%), 299.0755 (17.3%), ESI-Q-TOF HR-ESI-MS found: [M − H]^−^ = 298.0719 (100.0%), 299.0690 (23.9%), δ [ppm] = 1.2.

4-(4′-amino-[1,1′-biphenyl]-4-yl)-4-oxobutanoic acid **6o**



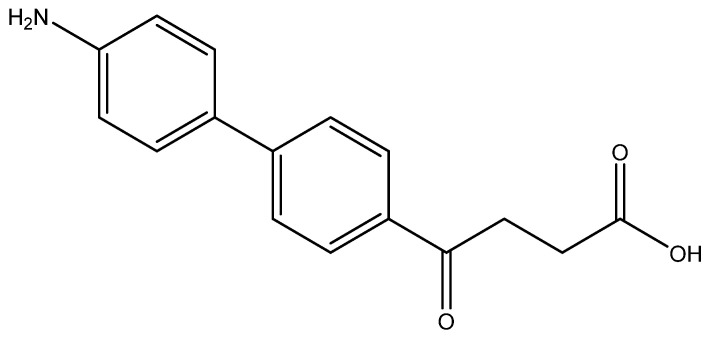



Reagents of **5o** (77 mg, 0.27 mmol), TFA (1.5 mL) and H_2_O (1.5 mL) were used. The procedure followed that described for preparing **6a**. Reaction was allowed for 1 h and TLC (CH_3_OH/CH_2_Cl_2_ = 1/19) indicated the consumption of the starting material (R*_f_* = 0.80) and formation of the product **6o** (R*_f_* = 0.10). After concentration under reduced pressure, the cooled CH_2_Cl_2_ was added. The mixture was filtered through suction. Additional cooled CH_2_Cl_2_ was added to wash the solid. The solid was then concentrated under reduced pressure to afford a cream-colored solid **6o** in 74% yield (53 mg, 0.20 mmol). m.p.: 150–153 °C.

**^1^H NMR** (500 MHz, CDCl_3_/CD_3_OD = 1:1) δ 2.72 (t, *J* = 6.5 Hz, 2H, CH_2_), 3.34 (t, *J* = 6.5 Hz, 2H, CH_2_), 7.14 (dd, *J* = 8.5, 2.5 Hz, 2H, aromatic), 7.61 (d, *J* = 8.5 Hz, 2H, aromatic), 7.66 (dd, *J* = 8.5, 1.5 Hz, 2H, aromatic), 8.02 (d, *J* = 8.0 Hz, 2H, aromatic); **^13^C NMR** (125 MHz, CDCl_3_/CD_3_OD = 1:1) δ 28.65 (CH_2_), 34.08 (CH_2_), 120.21 (CH, aromatic), 127.29 (CH, aromatic), 128.12 (CH, aromatic), 129.12 (CH, aromatic), 129.38 (CH, aromatic), 135.78 (C, aromatic), 145.23 (C, aromatic), 145.85 (C, aromatic), 176.06 (C, COO), 199.55 (C, CO); analysis for C_16_H_15_NO_3_, calculated [M − H]^−^ (*m*/*z*) = 268.0979 (100.0%), 269.1013 (17.3%), ESI-Q-TOF HR-ESI-MS found: [M − H]^−^ = 268.0960 (74.1%), 269.1041 (15.9%), δ [ppm] = −7.1.

methyl 4-(4′-((4-methylphenyl)sulfonamido)-[1,1′-biphenyl]-4-yl)-4-oxobutanoate **7**



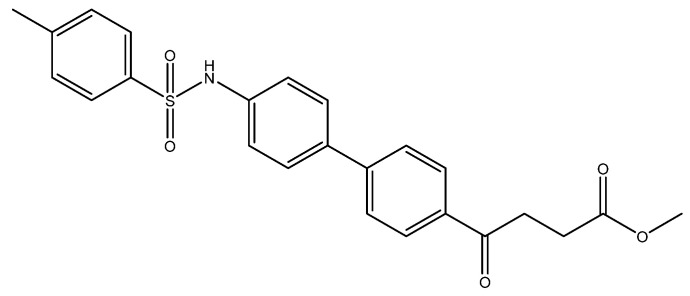



Compound **5o** (30 mg, 0.11 mmol, 1 eq) was co-distilled using toluene and CH_2_Cl_2_ three times followed by drying under high vacuo for 30 min. Dissolving by CH_2_Cl_2_ (0.5 mL) and adding pyridine (17 mg, 0.22 mmol, 2 eq), the mixture was cooled to 0 °C. *p*-TsCl (42 mg, 0.22 mmol, 2 eq) was added and the stirring was allowed for 10 min. Followed by removing the ice bath, the mixture turned from orange to pink during the next 30 min at rt. TLC (EtOAc/CH_2_Cl_2_ = 1/39) indicated the consumption of **5o** (R*_f_* = 0.4) and formation of the product **7** (R*_f_* = 0.32). After a further 17 h reaction, 1N HCl_(aq)_ (4.5 mL) was added followed by partitioning using CH_2_Cl_2_ (15 mL). Following washing with sat. NaCl_(aq)_ (3 mL) and collecting the organic layer, the aqueous layer was back extracted using CH_2_Cl_2_ (15 mL) twice. Combining the organic layers and drying over MgSO_4_, the mixture was filtered through gravitational filtration. The filtrate was concentrated under reduced pressure for subsequent flash chromatography using eluents (EtOAc/CH_2_Cl_2_ = 1:49) to afford white powder (88 mg). The analytic sample (58 mg) was crystallized from toluene to give white powder **7** in 48% yield (29 mg, 0.07 mmol). m.p.: 174–178 °C.

**^1^H NMR** (500 MHz, CDCl_3_) δ 2.46 (s, 3H, CH_3_, H_tosyl_), 2.78 (t, *J* = 6.5 Hz, 2H, CH_2_), 3.34 (t, *J* = 6.5 Hz, 2H, CH_2_), 3.70 (s, 3H, -OCH_3_), 6.76 (bs, 1H, -NH), 7.14 (d, *J* = 8.5 Hz, 2H, aromatic), 7.23 (d, *J* = 8.5 Hz, 2H, aromatic), 7.48 (d, *J* = 8.0 Hz, 2H, aromatic, H_tosyl_), 7.58 (d, *J* = 8.5 Hz, 2H, aromatic), 7.68 (d, *J* = 8.0 Hz, 2H, aromatic, H_tosyl_), 8.00 (d, *J* = 8.5 Hz, 2H, aromatic); **^13^C NMR** (125 MHz, CDCl_3_) δ 21.53 (CH_3_, C_tosyl_), 28.05 (CH_2_), 33.42 (CH_2_), 51.86 (CH_3_, O-CH_3_), 121.58 (CH, aromatic), 126.87 (CH, aromatic), 127.28 (CH, aromatic), 128.14 (CH, aromatic), 128.68 (CH, aromatic), 129.74 (CH, aromatic), 135.27 (C, aromatic), 136.21 (C, aromatic), 136.61 (C, aromatic), 136.75 (C, aromatic), 144.06 (C, aromatic), 144.69 (C, aromatic), 173.42 (C, COO), 197.54 (C, CO); analysis for C_24_H_23_NO_5_S, calculated [M + Na]^+^ (*m*/*z*) = 460.1189 (100.0%), 461.1223 (26.0%), ESI-Q-TOF HR-ESI-MS found: [M + Na]^+^ = 460.1187 (65.4%), 461.1216 (18.2%), δ [ppm] = −0.4.

4-(4′-((4-methylphenyl)sulfonamido)-[1,1′-biphenyl]-4-yl)-4-oxobutanoic acid **8**



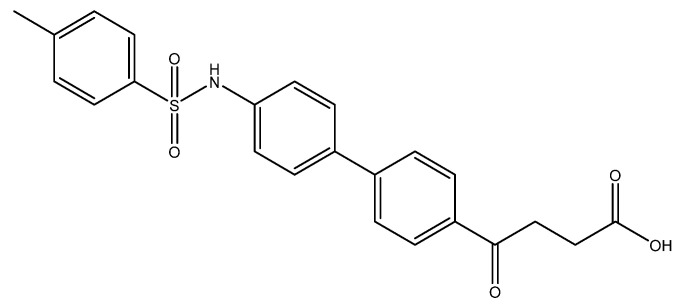



Reagents of **7** (17 mg, 0.04 mmol) and a solution of aqueous THF (0.6 mL, 75% *v*/*v*) were used. The gold solution was cooled at 0 °C and LiOH (2.9 mg, 0.12 mmol, 3.0 eq) was added. After 10 min, the ice bath was removed and the reaction was allowed for 1 h. TLC (CH_3_OH/CH_2_Cl_2_ = 1:19) indicated the consumption of the starting material **7** (R*_f_* = 0.84) and formation of the product **8** (R*_f_* = 0.30). It was treated with cationic exchange resin (strong H^+^). Followed by filtration and concentration of the filtrate, a white solid **8** was obtained in 75% yield (12.8 mg, 0.03 mmol). m.p.: 165–167 °C.

**^1^H NMR** (500 MHz, CD_3_OD/CDCl_3_ = 1:1) δ 2.33 (s, 3H, CH_3_, H_tosyl_), 2.71 (t, *J* = 6.5 Hz, 2H, CH_2_), 3.30 (t, *J* = 6.5 Hz, 2H, CH_2_), 7.17 (d, *J* = 8.5 Hz, 2H, aromatic), 7.22 (d, *J* = 8.0 Hz, 2H, aromatic, H_tosyl_), 7.47 (d, *J* = 8.5 Hz, 2H, aromatic), 7.59 (d, *J* = 8.5 Hz, 2H, aromatic), 7.66 (d, *J* = 8.0 Hz, 2H, H_tosyl_), 8.00 (d, *J* = 8.5 Hz, 2H, aromatic); **^13^C NMR** (125 MHz, CD_3_OD/CDCl_3_ = 1:1) δ 21.56 (CH_3_, C_tosyl_), 28.63 (CH_2_), 33.99 (CH_2_), 121.74 (CH, aromatic), 127.44 (CH, aromatic), 127.74 (CH, aromatic), 128.54 (CH, aromatic), 129.30 (CH, aromatic), 130.23 (CH, aromatic), 135.72 (C, aromatic), 136.43 (C, aromatic), 137.38 (C, aromatic), 138.63 (C, aromatic), 144.50 (C, aromatic), 145.88 (C, aromatic), 174.72 (C, COO), 199.35 (C, CO); analysis for C_23_H_21_NO_5_S, calculated [M − H]^−^ (*m*/*z*) = 422.1068 (100.0%), 423.1101 (24.9%), ESI-Q-TOF HR-ESI-MS found: [M − H]^−^ = 422.1070 (100.0%), 423.1098 (26.9%), δ [ppm] = 0.5.

*p*-hydroxymethyl fenbufen methyl ester **1****2**



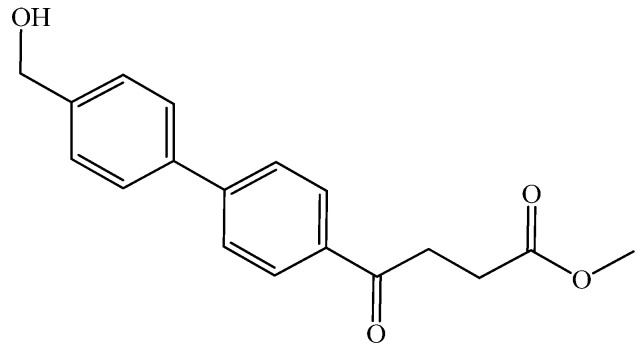



To a flask (250 mL) was added the bromo compound **2** (500 mg, 1.8 mmol, 1 eq) into THF (8 mL). The borono compound, 4-boronopinacol phenylmethanol (464 mg, 2.0 mmol, 1.1 eq), Pd(PPh_3_)_4_ (104 mg, 0.09 mmol, 0.05 eq) and aqueous K_2_CO_3_ (4.5 g, 32.4 mmol, 18 eq) in H_2_O (1 mL) were added, sequentially. The reaction was allowed at a gentle-reflux condition for 4 h (one drop per second). TLC (acetone/*n*-hexane = 3:7) indicated the consumption of the starting material (R*_f_* = 0.56) and formation of the product **12** (R*_f_* = 0.20). The mixture was partitioned between EtOAc (30 mL) and aqueous saline (satd., 15 mL × 3). The organic layer was collected, dried over Na_2_SO_4_, filtered through a celite pad. The filtrate was concentrated and chromatographed using eluents acetone/*n*-hexane in a gradient mode from 3:7 to 4:6 to afford a white solid which formed a transparent yellow solid under a high vacuum. The yield was 38% (205 mg). **^1^H-NMR** (500 MHz, CDCl_3_) δ 2.73 (t, *J* = 6.7 Hz, 2H, H_Aliphatic_), 3.21 (t, *J* = 6.7 Hz, 2H, H_Aliphatic_), 3.65 (s, 3H, H_OCH3_), 4.70 (s, 2H, CH_2_), 7.47 (d, *J* = 8.5 Hz, 2H, H_Ar_), 7.62 (d, *J* = 8.0 Hz, 2H, H_Ar_), 7.68 (d, *J* = 8.5 Hz, 2H, H_Ar_), 8.06 (d, *J* = 8.5 Hz, 2H, H_Ar_); ^13^C-NMR (125 MHz, CDCl_3_) δ 28.00 (aliphatic, CH_2_), 33.40 (aliphatic, CH_2_), 51.85 (OCH_3_, CH_3_), 64.84 (OH-CH_2_, CH_2_), 127.14 (Ar, CH), 127.38 (Ar, CH), 127.49 (Ar, CH), 128.45 (Ar, CH), 128.54 (Ar, CH), 128.63 (Ar, CH), 128.72, 131.95 (Ar, CH), 131.97 (Ar, CH), 132.02 (Ar, CH), 132.10 (Ar, CH), 135.21 (Ar, C), 139.06 (Ar, C), 141.09 (Ar, C), 145.51 (Ar, C), 173.42 (CO-OCH_3_, C), 197.65 (Ar-CO, C); analysis for C_18_H_18_O_4_, calculated [M + Na]^+^ (*m*/*z*) = 321.1 (100.0%), 322.1 (19.5%), 323.1 (1.8%), ESI+MS Q-TOF found: [M + Na]^+^ = 321.2 (7.01%), 322.2 (1.33%), m.p. 117–119 °C. Elem. anal. for C and H, calculated C: 72.47%, H: 6.08%, found C: 72.41%, H: 6.14%.

4-(4′-(hydroxymethyl)-[1,1′-biphenyl]-4-yl)-4-oxobutanoic acid **11**



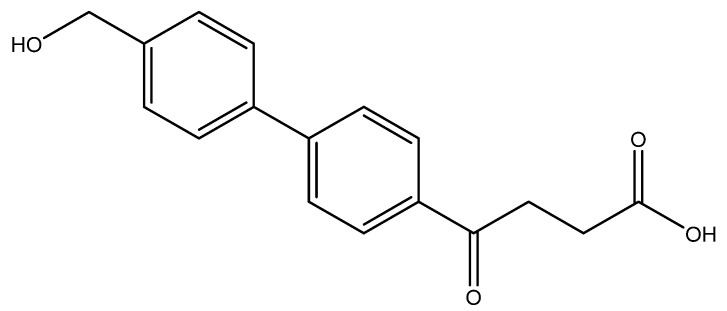



Reagents of **12** (72 mg, 0.24 mmol), TFA (3 mL) and H_2_O (3 mL) were used. The procedure followed that described for preparing **6a**. Reaction was allowed for 1 h and TLC (CH_3_OH/CH_2_Cl_2_ = 1/9) indicated the consumption of the starting material **12** (R*_f_* = 0.86) and formation of the product **11** (R*_f_* = 0.54). After column chromatography, a white solid **11** was obtained in 85% yield (58 mg, 0.2 mmol).

**^1^H NMR** (500 MHz, CDCl_3_/CD_3_OD = 1:1) δ 2.39 (t, *J* = 6.5 Hz, 2H, CH_2_), 2.96 (t, *J* = 6.5 Hz, 2H, CH_2_), 4.32 (s, 2H, CH_2_OH), 7.09 (d, *J* = 8.0 Hz, 2H, aromatic), 7.26 (d, *J* = 8.0 Hz, 2H, aromatic), 7.35 (d, *J* = 8.5 Hz, 2H, aromatic), 7.69 (d, *J* = 8.5 Hz, 2H, aromatic); **^13^C NMR** (125 MHz, CDCl_3_/CD_3_OD = 1:1) δ 28.74 (CH_2_), 34.15 (CH_2_), 64.52 (CH_2_), 127.72 (CH, aromatic), 127.78 (CH, aromatic), 128.09 (CH, aromatic), 129.27 (CH, aromatic), 136.05 (C, aromatic), 139.45 (C, aromatic), 142.32 (C, aromatic), 146.54 (C, aromatic), 176.00 (C, COO), 199.65 (C, CO); analysis for C_17_H_16_O_4_, calculated [M + H]^+^ (*m*/*z*) = 285.1121 (100.0%), 286.1155 (18.4%), [M + Na]^+^ (*m*/*z*) = 307.0941 (100.0%), 308.0974 (18.4%), ESI-Q-TOF HR-ESI-MS found: [M + H]^+^ (*m*/*z*) = 285.1124 (11.7%), 286.1154 (2.0%),δ [ppm] = 1.1, [M + Na]^+^ (*m*/*z*) = 307.0943 (12.5%), 308.0973 (2.1%), δ [ppm] = 0.7.

### 4.3. Bioassay

#### 4.3.1. IL-1beta Assay

Inhibition against the Il-1 beta expression level followed the same procedure that had been published previously by Hua GF and coworkers [50,51]. In brief, the mouse macrophage cell line J774A.1 purchased from the American Type Culture Collection (Rockville, MD) were primed for 5 h with 1 μg/mL lipopolysaccharide followed by treatment with the compounds in 50 μM and the standard MCC950 in 1 μM for 30 min. The mixture was then treated with 5 mM ATP for 0.5 h. The supernatants were then collected for assaying the IL-β using Elisa according to the brochure′s instruction—Invitrogen cat. No. 887013. The absorbance at λ_max_ of 450 nm were measured. The data were corrected by subtracting the absorbance at λ_max_ of 570 nm. Data were obtained in triplicate that had been corrected for the control group.

#### 4.3.2. COX-1 and COX-2 Inhibition Assay

Assay kit (No. 560131) from Cayman was used for screening bioactivity. The flowcharts have been appended as Appendix A. In brief, the volume of compound solution and the reagents used for this assay were increased from 10 to 20 μL in order that a multipipetting was performable. Ten microliters of Heme was diluted using 520 μL buffer solution. It was distributed to 7 tubes by 70 μL and the last tube by 40 μL. SnCl2 solution was prepared by dissolving 45 mg in HCl (0.9 mL) and distributing 120 μL to 8 tubes. The arachidonic acid solution was prepared by mixing a portion of 5 μL with 5 μL of KOH (aq) by vortex followed by dissolving with 1 mL H_2_O. It was distributed to 8 tubes in 125 μL amounts for each. A volume of 45 μL of COX enzyme dissolved in 415 μL buffer was ready for the subsequent procedure. In stage 2, all the reagents and compounds were added at volumes of 20 μL or 30 μL per multi pipetting, except the addition of COX enzymes. The last Elisa assay was performed by using the 2000-fold dilution and the addition stage was per multipipetting. The rest of development was the same as that described.

### 4.4. In Silico Modeling and Simulation

#### 4.4.1. Treatment before Docking Procedure

The sequence of enzymes COX-1 and COX-2 were retrieved from the two templates registered in PDB bank bearing the codes of 1EQG and 1CX2, respectively. They were both derived from the co-crystallization with the substrates of ibuprofen and bromocelecoxib, respectively. Ibuprofen is a COX-1 and COX-2 inhibitor with no selectivity. SC-558 (bromocelecoxib), like celecoxib, is a selective COX-2 inhibitor and is used as a template protein for present study. Before the molecular docking analysis, Discovery Studio’s (DS) Prepare Protein and Prepare Ligands were used to modify the charge distribution. The receptor part, i.e., the two COX enzymes, were both administered in the CHARMm force field throughout the whole docking process although the protein conformation seems not to be changed significantly. The flexible receptor atom property is enabled by creating a sphere radiating from a center defined by PDB crystal data in a radius of 4 Å as referred to the published work (Figure 11) [42,43]. The two sites contain mostly involved residues including Arg120 and Tyr355 for COX-1 and His90, Gln192, Arg513, Ser353, Tyr355 and Phe518 for COX-2. The sphere with a radius of 4 Å created in 1EQG covers additional residues, such as Val116, Arg120, Tyr348, Val349, Leu352, Ser353, Tyr355, Leu359, Phe381, Leu384, Tyr385, Trp387, Phe518, Met522, Ile523, Gly526, Ala527, Ser530 and Leu531; a volume for COX-2 (1CX2) encompasses residues of His90, Val116, Arg120, Gln192, Val349, Leu352, Ser353, Tyr355, Leu359, Phe381, Leu384, Tyr385, Trp387, Arg513, Ala516, Ile517, Phe518, Val523, Gly526, Ala527, Ser530 and Leu531.

#### 4.4.2. Algorithms and Scoring Functions

##### In Situ Ligand Minimization Algorithm

Available options for establishing the algorithm encompasses adopted basis Newton–Raphson (NR), steepest descent and conjugate gradient. NR is applied to a subspace of the coordinate vector spanned by the displacement coordinates of the last positions. In each step of this iterative procedure, the coordinates are adjusted in the negative direction of the gradient. Steepest descent does not generally converge, but will rapidly improve a very poor conformation. Conjugate gradient is an iterative method which makes use of the previous history of minimization steps as well as the current gradient to determine the next step. It has better convergence characteristics but is subject to numerical overflows when starting with very poor conformations. Energy minimization through these procedures will be scored using the smart minimizer function.

##### Conformation Method

Algorithm for generating conformations was enabled by adopting the option of FAST mode. This could provide rational numbers of low-energy conformation in a reasonable time.

##### Entropy Minimization

Similar to above in situ ligand minimization algorithm, the entropy component for the ligand conformation was also minimized using the tree approaches.

##### Implicit Solvent Model

Issues involved in this calculation regard the Coulomb repulsion and dielectric attraction. The option of Poisson–Boltzmann with non-polar surface area (PBSA) was adopted. PBSA is the most rigorous yet slowest solvent approximation method based on continuum electrostatics. This may not be available if in situ ligand minimization is running.

##### Implicit Solvent Dielectric Constant

The model used for dielectric constant for bulk solvent used is PBSA.

##### Salt Concentration

The solvation condition was generated through common settings, such as using NaCl as the salt and concentration was set to 0.145 M.

#### 4.4.3. Simulation Procedure

The docking simulation was performed using DS 2021 software integrated with the DS flexible docking protocol mode. The flexible algorithm allows the bound residue to fit in a reasonable manner. The pretreatment was the same as that described for the former docking simulation in Section 4.4.1. The solvation condition was generated through common settings, such as using NaCl as the salt and concentration in 0.145 M. The subsequent standard dynamics cascade protocol simulates the molecular dynamics through energy minimization and a number of stages, e.g., heating, equilibration and production. The procedures of steepest descent and conjugate gradient were used to converge as described above. The isothermal and isobaric ensemble conditions were set to perform the dynamics calculation. The candidate conformation was further analyzed in terms of trajectory based on the deviation from the initial atomic state which was described by two functions of root-mean-square deviation, RMSD, and root-mean-square fluctuation, RMSF.

The parameters were set to allow for maximum conformation numbers of ligand of 255, docking numbers of hotspot of 100 and energy threshold of 20. The probable conformations were submitted to DS Analyze Ligand Poses and Docking Pose in order to minimize binding free energy. A further validation using DS Calculate Binding Energies to compare the results from the two calculations can identify the most optimized conformation. All the free energy calculations will be enrolled in the following expression: ΔG_binding_ = ΔG_complex_ − ΔG_ligand_ − G_enzyme_.

The free energy calculation generates two classes of data expression: the preliminary binding energy of the system in the CHARMm condition and, additionally, the free energy resulting from the presence of solvent effect.

#### 4.4.4. Validation of Computational Program

The most optimal docking poses by the two benchmarking inhibitors, i.e., celecoxib and ibuprofen, were each superimposed with bromocelecoxib (SC58) and ibuprofen itself (Figure 12 and Figure 13). The latter two conformations were derived from the original co-crystallized complexes 1EQG and 1CX2, respectively. The two orientations of ibuprofen from the original complex and from simulation showed an RMSD value of 5.649 Å, a value relatively larger than the reported 0.433 Å [44]. The subtle shift in placement may be related to broader defining of the sphere covering the active site. When comparing bromocelecoxib (SC558) with celecoxib, an RMSD value of 6.615 Å is almost comparable to that of the ibuprofen group.

## Data Availability

Not applicable.

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
