# Peer review of "Antiinflammation Derived Suzuki-Coupled Fenbufens as COX-2 Inhibitors: Minilibrary Construction and Bioassay"

_molecules, 2022, doi:10.3390/molecules27092850_

Round 1

Reviewer 1 Report

The original article entitled “Antiinflammation derived Suzuki-coupled fenbufens as COX-2 inhibitors: minilibrary construction and bioassay” authored by Farn et al. describes the synthesis of fenbufen derived compounds and the inhibitory activity against cyclooxygenase-1 and -2 and Il-1beta. The manuscript presents a variety of novel compounds based on fenbufen as lead structure that were obtained from a versatile synthetic route. The manuscript is written clearly and in high detail.

The introduction gives a brief, clear and concise overview about COX with consideration on treatment strategies for SARS-COV-2 and a basis for principal dual targeting of COX and NLRP3 pathway. From this reviewers point it is recommended to summarize in a short paragraph the important structural key elements that are utilized to design COX and NLRP3 inhibitors which can serve as a basis for their development.

The results and discussion part covers the synthesis, which is described in high detail and delivered in an elegant way a set of different derivatives. The compounds were tested for their IL-1beta inhibitory action and showed lack of inhibitory potential for this target. In line with the COX inhibitory potential of fenbufen, the COX inhibitory potential was evaluated and the authors found COX inhibitory potency with selectivity for COX-2. The authors found considerable COX-2 inhibition at a concentration of 22 µM while COX-1 inhibition was low for all test compounds. The authors give a COX-2 vs COX-1 selectivity ratio/ratio which is defined in the COX-related literature but also in more general as ratio between two IC50 values (https://www.sciencedirect.com/topics/medicine-and-dentistry/selectivity-index). In this regard, this reviewer suggests to substitute the term selectivity ratio or selectivity index used in the manuscript with terms like ‘X-fold inhibition of COX-2 compared to COX-1’ or ‘X-fold higher COX-2 than COX-1 inhibition’. Docking studies gave further insights into the binding of COX. Has docking been performed for NLRP3 and can lack of inhibition be deduced from in silico studies as future guide for similar studies?

The experimental section is written in high detail and the identity of the novel compounds has been proven by adequate techniques. This reviewer acknowledges also the very precise description of steps to achieve absence of oxygen and moisture because, although in part standard techniques, it will help to reproduce these reactions in the future. The authors should furthermore report melting points for all the isolated solid compounds. Melting points have been reported only for two compounds yet but represent a fundamental parameter.

The conclusion gives some statements on further developments. In this regards the authors state that ‘analog 8 warrants a further study because of having a minor NLRP3 activity, a remarkable COX-2 activity’, which seems irritating because it was the aim of the study to identify NLRP3 and COX-2 inhibitor. Please clarify or rephrase this sentence. If possible, please shift the whole section conclusion to the end of results and discussion part and before the experimental part.

In conclusion, the paper should be considered for publication after minor revisions reported herein. With regards to the examination of their compounds also to Il-1beta and hence NLRP3 pathway, this reviewer emphasis the authors and editors to consider publishing this manuscript within the special issue “Theranostic Targeting of Cyclooxygenase-2: The Why, How and When?” also envisaged in molecules. From this reviewers point, this paper would finely fit to target one of the special issues aims to critically examine current developments and possibilities for application of selective COX-2 inhibitors.

Minor comments

Throughout the manuscript please change uM to  µM

27…Nonsteroid..Please change to Nonsteroidal

55.. Please consider rephrasing the sentence to: COX-1 and COX-2 are also less frequently named prostaglandin endoperoxide synthase-1 and -2 (PGHS-1 and -2) in the literature.

Figure 8-11. Please change ‘substrate’ to ‘compound’ since investigated compounds are not converted as substrates by COX

88… Please consider rephrasing: ‘Fenbufen analog as a typical starting material’ to ‘Fenbufen analog as lead compound of this study’

103.. please check ‘boronic ester 5’. Shouldn’t it be ‘boronic ester 3’?

105.. Please check. Is really acetone formed as byproduct of cleavage or pinacol?

177… Please replace ‘elisa staining’ with ‘detection by an enzyme-linked immune assay (ELISA)’

198… Please change ‘comparable COX-2 activities to that of celecoxib’ to ‘comparable COX-2 inhibition to that of celecoxib’ to

299 ‘Angilent’ to ‘Agilent’

300 ‘UVIS’ to ‘UV-Vis’

399 ‘with1’ to ‘with 1’

543… ‘potion’ to ‘portion’

972.. ‘crystalized’ to ‘crystallized’

1068… ‘ina’ to ‘in a’

1069..’ 2000-flod dilution version’ to ‘2000-fold dilution’.

Author Response

Dear Prof. reviewer,

Thank for your thorough reviewing my work. The corrections have been made and marked in red. Please feel free to let me know whether any corrections are needed to be addressed. 

Sincerely,

Chung-Shan Yu

Reviewer 2 Report

The manuscript (molecules-1662876) entitled " Antiinflammation derived Suzuki-coupled fenbufens as COX-2 inhibitors: mini-library construction and bioassay” by Chung-Shan Yu and co-worker "describes a series of 18 molecules based on fenbufens which are evaluated for their anti-inflammatory activity against IL-1B and for their ability to inhibit cyclooxygenase. An additional computational study is also provided. The abstract is well-written and accurately describes the overall study. However, several concerns must be addressed before the manuscript is accepted for publication.

  1. MD Discussion part is completely missing from the computational part of the manuscript. Are the free energies of binding computed from MD simulations favorable for binding? Are these energies comparable to those obtained for a known benchmarking inhibitor?
  2. This manuscript lacks a strong conclusion. A detailed conclusion should be added. A brief description of Author’s findings should be added to the manuscript during revision.
  3. The general quality of the figures can be improved.
  4. In the discussion section, the authors should outline some details about the docking protocol e.g. the docking algorithm, the score function , …..
  5. Please, Modify the 3D images in the docking study so that all the amino acids appear clearly.
  6. What force-field parameters did the authors use for ligands in MD simulations since they are non-standard compounds and not readily available in any MD package?
  7. The authors should comment on how many poses were obtained by the docking simulations and how the chosen poses validate the activity.
  8. It is unclear do the authors performed energy minimization of the protein structure before docking. The description should be added.
  9. In the discussion section, the authors should outline the limitations of the docking algorithm used in the study and discuss how those limitations will affect their predictions.
  10. In the methods section, the authors should summarize the docking algorithm and especially the scoring function and the docking simulation protocol, and provide references describing the benchmarks that were performed to validate the docking program.
  11. Please superimpose the most active compound and the crystallized ligand for comparison and explain the binding mode.
  12. While the docking studies are interesting, the various text and figures take a significant amount of journal space and likely could be summarized more concisely. One suggestion is to show data and figures for the docking of just one compound (like 6a) in the main body of the manuscript and move any discussion and figures for the docking of other compounds to the supporting information file.
  13. A correlation of binding affinity, selectivity, and conformational changes could be discussed.
  14. In the modeling section, it is assumed that 1EQG and 1CX2 are the targets, but I could not get any rationale behind this. The authors should better explain why they make this assumption.
  15. The text in the manuscript's main body, should be shortened and further focused for clarity and ease of readability.

Author Response

Dear reviewer,

I thank you for the thorough reviewing wotk. The correction has been made accordingly. Please feel free to point out any mistakes for further improvement.

Sincerely,

Chung-Shan Yu

Round 2

Reviewer 2 Report

The revised version of the manuscript of Chung-Shan Yu and co-workers has been substantially improved. The authors implemented all of the suggested corrections. I accept this manuscript in its present form.